# HypE-GA based study on optimal design of standard floor facade windowing of high-rise office buildings facing energy saving in heating, cooling and lighting

Weixiang Zhang[1]*, Jieli Sui[2]*

1 School of Architecture and Engineering, Yantai Institute of Technology, Yantai, Shangdong, China,
2 School of Architecture, Yantai University, Yantai, Shangdong, China

\* zhangweixiang2022@126.com (WZ); suijieli@126.com (JS)

**Data Availability Statement:** All relevant data are within the manuscript and its Supporting Information files.

## Abstract

The quantitative design on area and location of building façade's windows has a significant impact on interior light and heat environment, which is also very instructive for preliminary and remodeling design of buildings. However, previous studies paid more attention to the thermal insulation construction and shading based on design parameters from the perspective of designers, but neglected the fact that the geometric properties of the windows themselves are equally important for building energy efficiency. Secondly, the weak interactivity and algorithmic limitations of traditional simulation platforms prevent rapid access to ideal design strategies. Therefore, this paper takes the standard floor of a high-rise office building as the research object in cold region−Yantai, facing façade windowing design, the three building performance objectives of each office unit−Annual Cooling Energy Consumption (AC), Annual Heating Energy Consumption (AH) and Annual Lighting Energy Consumption (AL)−are simulated and single/multi-objective optimized by relying on Ladybug and Honeybee (LB + HB) platform and Hypervolume Estimation Genetic Algorithm (HypE-GA) to obtain the genome of Pareto−Window-to-Wall Ratio (WWR), Window Height (WH) and Sill Height (SH)−at the lowest of each performance objective in order to determine the most energy-efficient façade windowing expression. The results show that AH and AC, their sum of quantities remains stable, are main energy consumption sources of office buildings, while the change of AL is more likely to have an impact than the others' on Annual Totaling Energy Consumption (AT). The analysis points out that different windowing strategies can be adopted for different performance objectives. To reduce AC, priority is given to windowing on the east and north facade, with East Window-to-Wall Ratio ($WWR_E$) at 0.2 ~ 0.3 and North Window-to-Wall Ratio ($WWR_N$) at 0.3 ~ 0.5; to reduce AH, windows on the west and north facade should not be opened, and the remaining facades should be opened in small areas; to reduce AL, WWR> 0.7 is appropriate for each facade, and should be considered to matching a higher SH or WH; From AT, the average WWR in the single-objective and multi-objective optimization results are similar, so it is suggested that the WWR of each facade of office buildings in Yantai area is $WWR_E$ = 0.47, North South Window-to-Wall Ratio ($WWR_S$) = 0.46, West Window-to-Wall Ratio ($WWR_W$) = 0.18 and $WWR_N$ = 0.54. In addition, this

**Funding:** The research described in this paper was funded by Yantai Institute of Technology Youth Foundation Program [2023QNJJ03] and Yantai Institute of Technology Academic Team Program [2022XSTD04] in the form of grants to WZ, and by Shandong Social Science Planning Research Program [22CWYJ35] in the form of a grant to JS.

**Competing interests:** The authors declare that they have no known competing financial interests or personal relationships that could have appeared to influence the work reported in this paper.

paper proposes a method that can quickly find the Pareto optimal solution by clustering analysis on optimized results through Origin in multi-objective HypE-GA optimization study.

## 1 Introduction

In the 21st century, buildings remain the largest part of global energy consumption, accounting for 40% of total global energy consumption and one-third of greenhouse gas emission [1]. If countries do not improve energy efficiency in building sector, it is impossible to achieve their emission reduction [2]. China has proposed the strategy of "2030 carbon peak and 2060 carbon neutral", actively echoing the Paris Agreement's goal of achieving 2°C temperature control. China's construction industry accounts for more than 30% of the world's primary energy consumption and generates 27.9% ~ 34.3% of carbon emission [3, 4]. In 2021, buildings' total energy consumption reached 19.1 tec, accounting for 36.3% of national energy consumption (13.8% for materials' producing, 0.6% for constructing and 21.9% for operating), and the total carbon emission was 4.07 billion tCO2, accounting for 38.2% of national carbon emission (16.0% for materials' producing, 0.6% for constructing and 21.6% for operating). From 2010 to 2021, Chinese public building area has an 91% increase dramatically, from 7.8 billion $m^2$ to 15.0 billion $m^2$ [5]. As public buildings, office buildings have a greater impact on the energy environment that are second only to residential buildings and shopping mall buildings in terms of energy consumption quantity and Energy Use Intensity (EUI), respectively [6]. In the operational phase, the air conditioning system accounts for 40% ~ 50% and lighting system for 30% ~ 40% in energy consumption, respectively [7], so it is crucial to reduce energy consumption in these aspects.

Exterior windows are the key part of light and heat transfer through the indoor and outdoor, and more than 50% heat loss caused by them in the entire enclosure system [8], potentially increasing the energy load. Chinese office buildings in cold regions are not well designed in their envelope structure, resulting in their regrettable performance in terms of building energy efficiency [3]. Design parameters such as Window-to-Wall Ratio (WWR), Window Height (WH) and Sill Height (SH), orientation and shape of windows play a decisive role in Annual Cooling Energy Consumption (AC), Annual Heating Energy Consumption (AH), Annual Lighting Energy Consumption (AL) and Annual Totaling Energy Consumption (AT) [9, 10], so designers should first seek breakthroughs in windowing design, and then achieve the building energy saving in the operational phase. The energy consumption quantities counted in the simulation study in this paper come from the equipment terminal and do not consider other losses. The AH and AC are from the HVAC system terminals and the AL is from the lighting system terminals.

In the early design stage, building performance simulation evaluation is much more effective than post-use evaluation [11, 12]. Regarding the design of energy-efficient windowing, scholars have done a lot of research, which can be broadly divided into two categories. The first category is based on building performance objectives to study the impacts of windows' design parameters [13–17], while analyzing buildings' performances in different climate zones and finding that appropriate windowing design has a positive effect on energy efficiency, indoor comfort, and daylighting [18, 19]. To maintain the light-heat balance, elements related window geometric properties were shown to be the most influential class on building performances from a study on façade windowing optimal design for a high-rise office building in a tropical region [10, 20].

Previous scholars have often selected solutions by performing sensitivity analysis on design parameters [21–23], but two drawbacks are obvious: (1) the sensitive weights of design

parameters are not assigned precisely enough; (2) the base group of design solutions expands exponentially as the dual increase of design parameters' number and performance indicators' dimension, thus consuming a lot of time and effort. In two earlier studies [24, 25], two parts of the work were specifically done. Not only verified that window geometric parameters have the greatest sensitivity to building performance, but also remedied the above two research short-comings. In the first part, the multicollinearity problem between high-dimensional design parameters and multi-objective building performance (building energy consumption, day-lighting, indoor thermal comfort) was solved by principal component analysis. It was found that only 12 of the preset 15 parameters passed the KMO Sampling Suitability Quantity and BARTLETT sphericity test, and they were classified into four principal components: Building Body (Bay, Depth and WWR), Windowing Geometry (Window Height, Sill Height and Win-dow Horizontal Distance), Window Properties (K-value, Solar Heat Gain Coefficient and Visual Transmittance), and Facade Properties (Wall Height, Plane Corner, External Wall R-Value), and the ensuing weight calculations yielded that the weights of each principal component were 0.41, 0.268, 0.188, and 0.134 in that order, thus proposing the principal component linear function assessment model. In the design of buildings with Bay and Depth, the Windowing Geometry with WWR is the most sensitive type of design parameters for building performance [24]. In the second part, the sensitivity of window geometric parameters to building energy consumption (AC, AH, AL and AT) was investigated by variance calculation. It is shown that WWR is the dominant factor affecting indoor AC and AH in the design of facade windowing, and larger values of WWR, WH and SH within a certain range are preferable in terms of AT reduction. The best orientation for window-opening is south, east, north, and west in a sequence when WWR is in the range of 0.3 to 0.7. Different factors have varied influences on AT under different orientations: $WWR_S > WWR_N > WWR_E > WWR_W$; $WH_E > WH_S > WH_N > WH_W$; $SH_S > SH_E > SH_N > SH_W$ [25].

The second category is based on algorithms to obtain windowing strategies quickly. With the help of information technology platforms such as EnergyPlus, Grosshopper, Design-Builder, DeST series etc. [26–29], supplemented by intelligent algorithms such as CA (Cellular Automata), GA (Genetic Algorithm), ANN (Artificial Neural Network), DNN (Deep Neural Networks) etc. to drive building model simulation and operation [30–33], the final analysis the correlation and sensitivity of window design parameters and performance metrics. Compared with the first category, this type can provide both strategic analysis and optimal design of win-dowing under multi-objectives study. In recent years, parametric design tools have progressed in integrating simulation techniques, among which Ladybug and Honeybee (LB + HB) parametric energy-efficient design is particularly prominent and has been more frequently applied in building performance studies [34–36]. LB + HB can be combined with genetic algo-rithms (GA) to quickly realize the interaction and iteration of input layer and output layer, solving the lacks of precision and calculation in the previous optimal design. In addition, Phil-lips et al. [20] studied the connection between window properties and building performances in office buildings under different climate zones, emphasizing the necessity of multi-objective windowing design and evaluation, instead of alone relying on single-objective optimization to make decisions.

Nowadays, the green concept in green building, urban renewal and remodeling has attracted much attention. In previous architectural research and practice, there are numerous software for simulating building energy efficiency, but their single functionality and weak interaction between software cannot fully meet the design needs of architects. Therefore, on the one hand, to solve the problems of unfriendly multi-platform interactivity and low optimi-zation efficiency, especially for design projects based on multi-objective optimization. On the other hand, to provide architects with objective and fast decision support for early design and

later remodeling of façade windowing. Studying the relationship between windowing design and single/multi-objective building performances in high-rise office buildings in cold region applying through LB + HB and Hypervolume Estimation Genetic Algorithm (HypE-GA), which becomes an innovative aspect that differs from above studies.

The paper aims to propose a more complete optimal design approach oriented towards maximizing energy efficiency, and then quantitatively guiding architects in windowing design. The research focuses on single/multi-objective optimization of building performances by HypE-GA and finding Pareto or Pareto set solutions quickly. The research structure of this paper is as follows: (1) Section 1 introduces the research background, existing problems, research objectives and solutions. (2) Section 2 mainly describes the research method systematically, including the research system boundary, calculation models, optimization algorithm and functions of research tools. (3) Section 3 establishes the energy model and sets up the simulation environment; (4) Section 4 conducts single/multi-objective optimization and clustering to find the optimal solution. (5) Section 5 gives the conclusions and future work of this research.

## 2 Methodology

### 2.1 System boundary

Based on HypE-GA and LB + HB simulation platforms, a standard floor of a tower-type high-rise office building in Yantai, a cold region of China, is selected as the research object. The four aspects of energy consumption, AC, AH, AL and AT, are used as observation points to explore the windowing positions under single-objective/multi-objective building performances optimization, and to obtain genomes of WWR, WH, SH, Window Width (WW) and Window Edge Distance (WED), where WWR, WH, and SH are primary design parameters and WW and WED are secondary design parameters, derived from primary design parameters according to horizontal equipartition principle.

### 2.2 Calculation models

The calculation models for single-objective and multi-objective optimization of this paper, Eq (1) and Eq (2), respectively, are as follows:

$$F_1(X) = \min(f_n(X)) = [a_n \ \ b_n \ \ c_n \ \ C][x_1 \ \ x_2 \ \ x_3]^\top \tag{1}$$

$$F_2(X) = \min(f_1(X) \ \ f_2(X) \ \ f_3(X)) = \begin{bmatrix} a_1 & b_1 & c_1 & C_1 \\ a_2 & b_2 & c_2 & C_2 \\ a_3 & b_3 & c_3 & C_3 \end{bmatrix}[x_1 \ \ x_2 \ \ x_3]^\top \tag{2}$$

Where, $n$ takes 1, 2 or 3, $X = [x_1 \ \ x_2 \ \ x_3]^\top = [WWR \ \ WH \ \ SH]^\top$, $a_n, b_n, c_n$ are weighting factors of $WWR \ WH$, $SH$ respectively, $C$ is a constant. The range of values for each design parameter (WWR, WH and SH) is weighted. In Eq (1), the single-objective optimization process is infinitely close to a unique optimal solution, and in Eq (2), multiple optimal solutions will occur in multi-objective coupled optimization.

Where in Eq (1), AC, AH and AL are executed for single objective optimization respectively. As below:

$$f_1(X) = AC(WWR \ \ WH \ \ SH) \rightarrow \min(EUI),$$

$$f_2(X) = AH(WWR \ \ WH \ \ SH) \rightarrow \min(EUI),$$

$$f_3(X) = \mathrm{AL}(WWR\ WH\ SH) \rightarrow \min(EUI).$$

Where in Eq (2), AC,AH and AL are executed simultaneously for multi-objective optimization. As below:

$$\begin{aligned} F_2(X) &= f_1(X) + f_2(X) + f_3(X) \\ &= \{\mathrm{AC}(WWR\ WH\ SH) + \mathrm{AH}(WWR\ WH\ SH) + \mathrm{AL}(WWR\ WH\ SH)\} \rightarrow \min(EUI) \end{aligned}$$

## 2.3 Optimization algorithm

GA is a randomized adaptive search algorithm based on evolutionism, which treats design variables as genes and imitates the genetic mechanism of biological "selection", "crossover" and "mutation". Finally, through the fitness function, it converges continuously throughout the search domain and seeks the optimal solution or infinitely close to it, i.e., the Pareto or Pareto front in economics. For solving multi-parameter and high-dimensional optimizing, GA is more accurate than other intelligent algorithms and improves simulation efficiency by 50% [37]. Compared to eight algorithms such as particle swarm optimization and gradient algorithm, GA is favored by researchers for its higher accuracy and exponentially reduced search time [38]. GA is also widely used in architectural design [39], especially in building performances optimization. Hosamo et al. [40] used a hybrid technique of GA and least squares support vector machine to improve building energy consumption and thermal comfort in his study. Khalil A. et al. [41] studied buildings' energy-use density in three different climate zones to optimize building form with the help of GA. Jalali Z. et al. [42] combined construction projects to obtain Pareto solutions for energy-efficient design of building envelope and indoor light-heat environment, starting from reducing operational energy consumption. Junchao H. et al. [43] found the optimal windowing strategy with the help of GA, considering building energy efficiency and visual comfort. In this study, Octopus on Grasshopper platform, a GA plugin, was used to iterate to find the best solution that satisfies the objective optimization. The optimization was performed using HypE-GA embedded in Octopus. HypE-GA uses Monte Carlo simulations to approximate the exact hypervolume values. The main idea is that the actual metric value is not important, but the ranking of solutions induced by the hypervolume metric is important. Thus, hypervolume-based search not only solves multi-objective problems, but also flexibly adjusts the running time. Bader and Zitzler showed HypE-GA was used for more accuracy and speed than similar algorithms, including Strength Pareto Evolutionary Algorithm II (SPEA-II), Nondominated Sorting Genetic Algorithm II (NSGA-II) and Indicator-Based Evolutionary Algorithm (IBEA), in the optimization process in cases with more than three fitness functions [44]. The HypE-GA was matched with HypE Reduction convergence mechanism and HypE Mutation mechanism. The set of objective value vectors corresponding to each solution in the Pareto Set is called the Pareto Front. Hype Reduction stipulated how to truncate a Pareto front solutions to fit the result size when it is too big. HypE Mutation applied mutation and recombination operators to generate N offspring to simulate the diversity of iterative processes. Therefore, HypE-GA is an appropriate way to solve single-objective or multi-objective optimization problems in this study, and Fig 1 shows HypE-GA based optimization mechanism of building performances.

The operating parameters, including Elitism, Mutation Rate, Mut.Probability, Crossover Rate, Population Size and Max Generations, are very critical because different settings of parameters will have a large impact on the efficiency of HypE-GA. Standard GA discard individuals from the previous generation at each iteration. To better at retaining elite individuals here Elitism gives at least 50% new solutions that are bred out of the Elite instead of the entire pool. The lower Crossover Rate, the less information is exchanged between individuals,

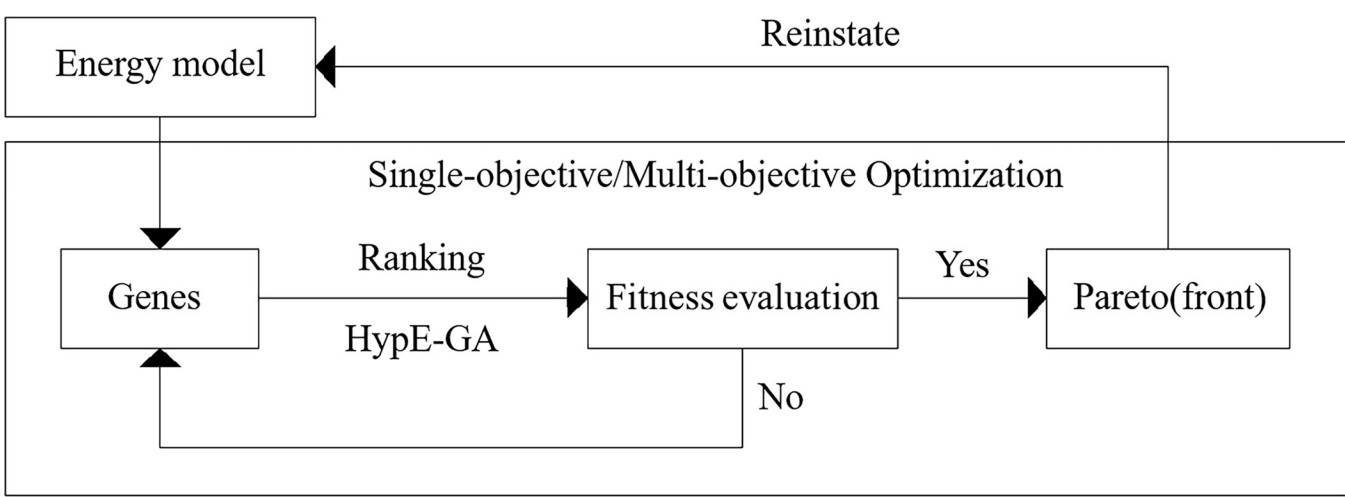

**Fig 1. Optimization mechanism of building performances.**

resulting in undercharacterised new solutions, so a value of 0.4 ~ 0.99 is recommended. A low Mutation Rate means little changes to the parameters' values, multiplied by Mut.Probability is generally recommended to take a value of 0.0001 ~ 0.1. Population Size and Max Generations directly affects the efficiency of HypE-GA, which can be set a value according to the case complexity in 10 ~ 50 and in 10 ~ 200, respectively. In this case study, based on the above principles, operating parameters of single-objective HypE-GA are shown in Table 1, and that of multi-objective HypE-GA are shown in Table 2.

## 2.4 Research tools

Fig 2 shows the introduction of tools used in the research process. According to the research idea can be divided into four stages: (1) Parametric model. Grosshopper is a graphical programming language for Rhino, which allows researchers to freely determine design parameters and ranges to build models, and allows easy interface and data visualization management, and is the most popular parametric design tool in the industry [45]. (2) Energy consumption simulation. LB+HB is the open-source code of grosshopper, written in python language, which can realize graphical language interaction in Rhino interface, and can realize energy consumption simulation while managing model programs and image data. LB can read meteorological data (from Energyplus website). The LB + HB platform uses four compatible plug-ins, Daysim (V4.0.0), Radiance (V 5.0.a.6), Energyplus (V9.0.0) and Openstudio (V2.5.0), which can be used according to different design needs. Among them, Daysim with better simulation accuracy, the same computing kernel as Radiance, is meteorological data based to dynamically simulate year-round daylighting [46]. Compared with Energyplus, Openstudio simplifies many modeling procedures and calculations, and allows fast option of space types and HVAC systems. (3) HypE-GA optimization. A plug-in Octopus (V4.0) with HypE-GA, which connects design variables at the input (G port) and building performance metrics as evaluation targets at the output (O port), drives the energy model with iterative operations (see Fig 3). Double-click G to expand the Octopus operator, which is divided into four regions, R1~R4. R2 is the

**Table 1. Single-objective HypE-GA assignment.**

| Sub-items | Elitism | Mutation Rate | Mut.Probability | Crossover Rate | Population Size | Max Generations |
|---|---|---|---|---|---|---|
| Settings | 0.5 | 0.01 | 0.2 | 0.6 | 30 | 20 |

**Table 2. Multi-objective HypE-GA assignment.**

| Sub-items | Elitism | Mutation Rate | Mut.Probability | Crossover Rate | Population Size | Max Generations |
|---|---|---|---|---|---|---|
| Settings | 0.5 | 0.01 | 0.2 | 0.5 | 50 | 100 |

main view region, and the R1 region has two functions, one is to regulate the size and the state of the main view and the solution, and the other is to observe the state of each iteration, such as calculating once time, the number of dominated and undominated, and the size of the evolutionary pool. R3 region is to observe the convergence degree, parameter diversity and coordinate naming of performance goals. R4 region is to set the parameters of HyeE-GA and select the convergence mechanism, and mutation mechanism. (4) Data analysis. Import calculation results into Origin (2017), and draw a 3D scatter plot when conducting multi-objective optimization studies, and then obtain Pareto or Pareto set solutions through simple clustering analysis.

## 3 Case study

### 3.1 Energy model creation

A tower-type office building with north-south orientation, frame-core construction, the standard floor of which was selected as the object model-ruled, and divided into nine same units, each size for 12 m × 9 m × 4 m. Among them, eight office units are distributed around the traffic core notated as office1 ~ office8 (see Fig 4), representing the eight directions of office units in North (N), Northwest (WN), West (W), Southwest (SW), South (S), Southeast (SE), East (E) and Northeast (NE). WWR, WH and SH, i.e., genes, their value ranges are shown in Table 3. Horizontally, the WW is the width of the window opening and the WED is the distance from the side of the window opening to the demarcation line of the office unit (see Fig 4). While WW and WED can be extrapolated from WWR, WH and SH, to introduce Eq (3) and Eq (4), respectively:

$$WW = 4 \times L \times \frac{WWR}{WH} \tag{3}$$

$$WED = \frac{(L - WW)}{2} \tag{4}$$

Where, L is the width of Bay.

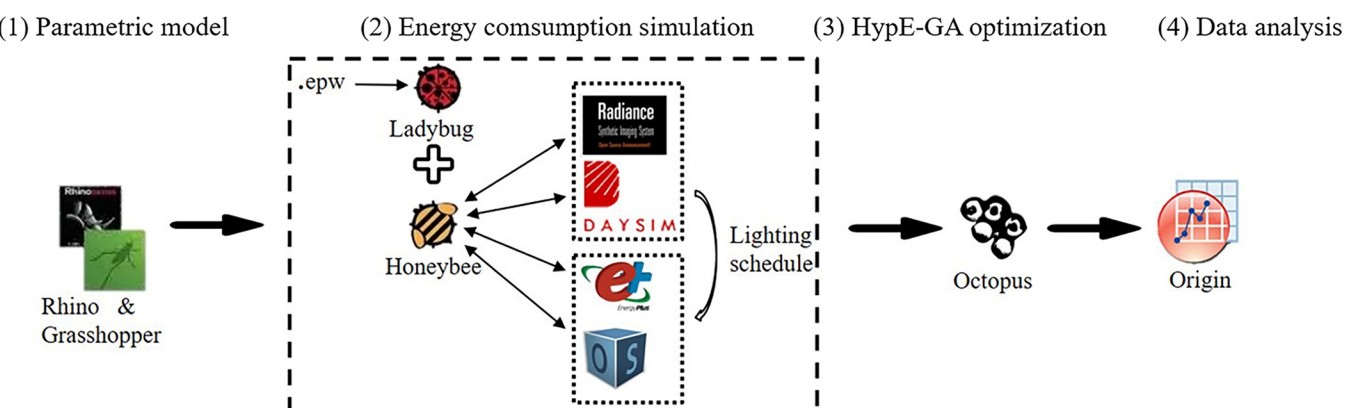

**(1) Parametric model**   **(2) Energy comsumption simulation**   **(3) HypE-GA optimization**   **(4) Data analysis**

**Fig 2. Tools applied to the research process.**

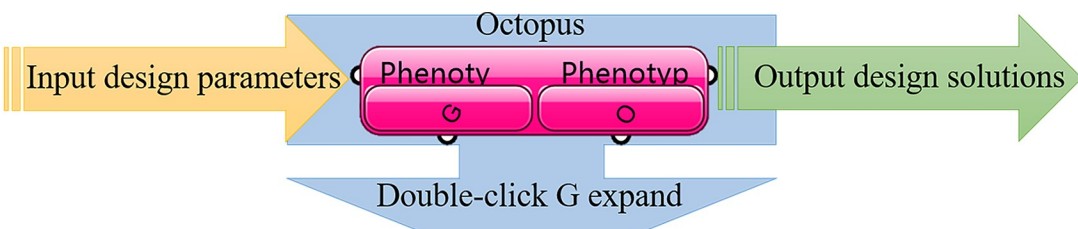

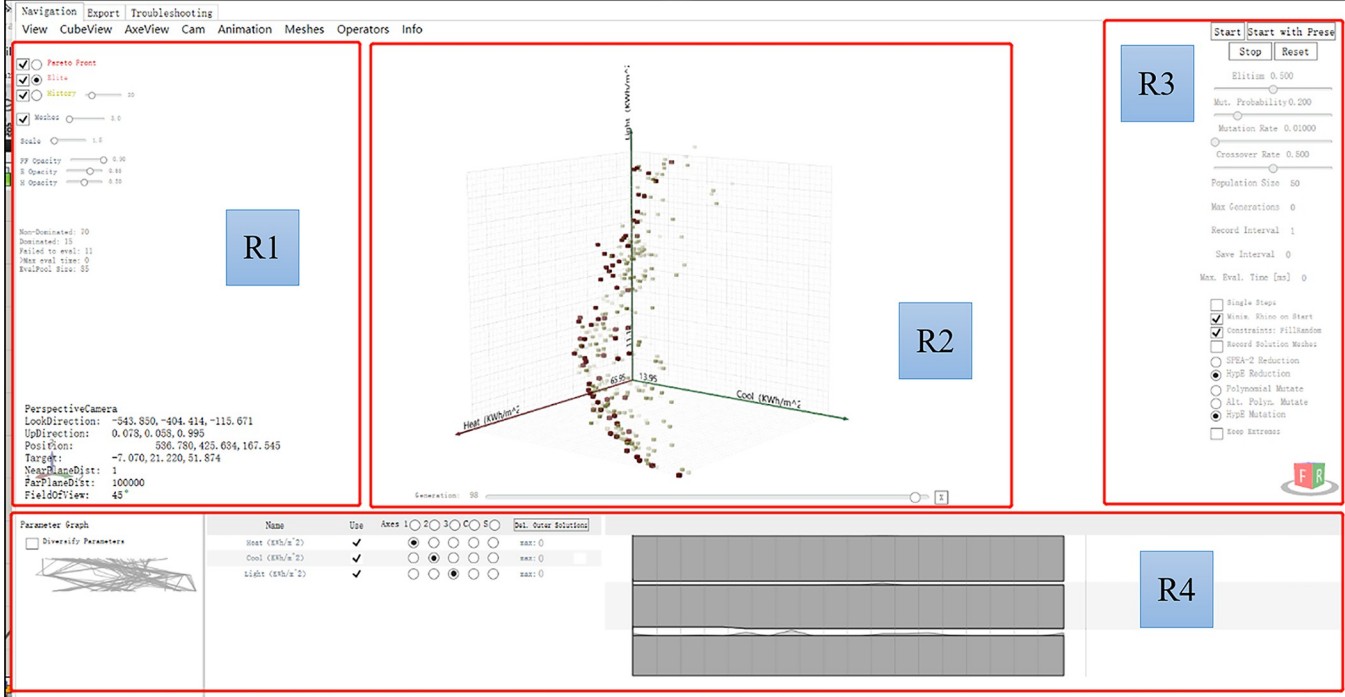

**Fig 3. Octopus operator interface composition.**

### 3.2 Simulation environment setting

**3.2.1 Meteorological conditions setting.** The typical meteorological year (2005) data of Yantai Longkou weather station were read by LB. This is a Chinese Standard Weather Data (CSWD) researched and developed by the Meteorological Information Center of the China Meteorological Administration in conjunction with the Department of Building Science of Tsinghua University, which was updated in 2005 and is still commonly used in studies of building simulation. In addition, in 2005, the Yantai region was hit by a huge snowstorm, and the lowest temperature broke the record within 50 years, which was called the year of extremely low temperature, so the meteorological data of that year is particularly important for studying energy conservation in heating demand buildings in northern China. The meteorological data for recent years (2006 to 2022) have been relatively stable compared to 2005, and are therefore not as representative as they could be. For heating buildings in the current northern cold zone, it is necessary to consider the historical extreme cold years as the meteorological conditions for simulation.

**3.2.2 Materials creation.** For components of the standard floor envelope, a common building construction practice was selected. Table 4 shows the materials used in building enclosure and their physical properties.

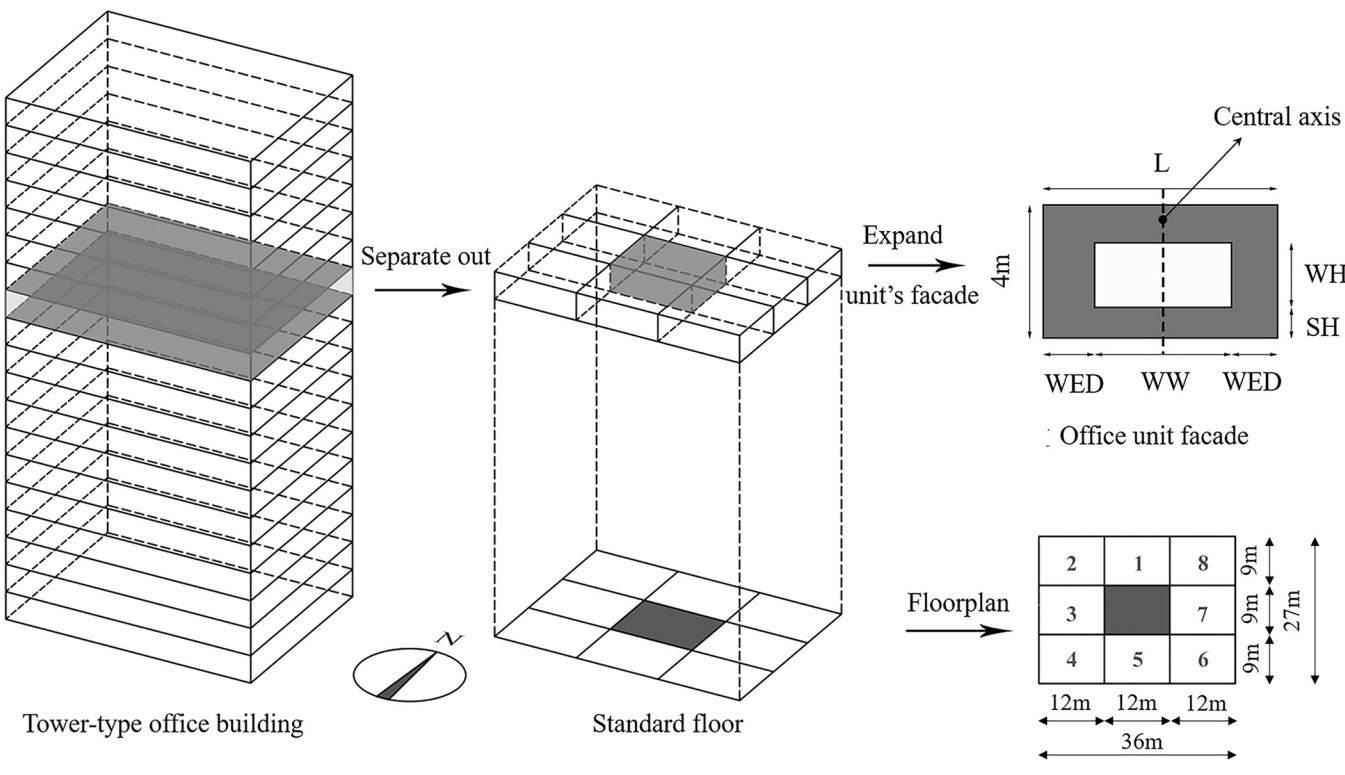

**Fig 4. Standard floor office unit model.**

**3.2.3 Indoor settings.** According to Design Standard for Energy Efficiency of Public Buildings GB50189-2015 [7], the office building operating hours are 7:00–18:00 on weekdays and holidays are set to weekends, and some load settings are shown in Table 5 based on GB50189-2015. Fig 5 shows the operation schedules of indoor cooling, heating, lighting, occupancy rate, electrical equipment utilization rate and fresh air operation. Among them, the lighting and electrical equipment utilization rate schedule is the same, and the fresh air system is always on. According to Method of Daylighting Measurements GB/T5699-2017 [47], a working surface 0.75 m high from the indoor floor and a network of 2 m × 2 m light test points are set up in the Radiance component. Lighting is used as a compensatory measure for indoor daylighting, and the lighting time obtained from dynamic lighting simulation is used to replace the lighting time in Fig 5, and artificial lighting will be automatically turned on for illuminance levels below 300 lx.

**3.2.4 HVAC setting.** The most widely HVAC system of combination of VRFs (Variable Refrigerant Flow system) +DOAs (Dedicated Outdoor Air system) in office buildings is used in this study, and then quickly configure the parameters of VRFs + DOAs in the Openstudio component (see Table 6).

**Table 3. Design parameters and their range.**

| Design parameters | Range | Step length |
|---|---|---|
| WWR | 0 ~ 0.95 | 0.01 |
| WH | 0 ~ 4.0 m | 0.1m |
| SH | 0 ~ 4.0 m | 0.1m |

**Table 4. Enclosure structure construction and thermophysical performance parameters.**

| Constuction name | Main Material | Thickness (m) | Thermal Conductivity (W/m·K) | Dry Density (kg/m$^3$) | Specific Heat Capacity (J/kg·K) | Thermal Absorptivity | Solar/Visible Light Absorptivity |
|---|---|---|---|---|---|---|---|
| Roof/Floor | Marble | 0.02 | 2.91 | 2800 | 920 | 0 | 0.6 |
| | Lime Cement Mortar | 0.03 | 0.87 | 1700 | 1050 | 0 | 0.48 |
| | Reinforced Concrete | 0.15 | 1.74 | 2500 | 920 | 0 | 0.73 |
| Interior Wall | Lime Cement Mortar | 0.02 | 0.87 | 1700 | 1050 | 0.9 | 0.48 |
| | Stucco Plaster Pressed into the Mesh Cloth | 0.005 | 0.29 | 800 | 1050 | 0.9 | 0.68 |
| | Thermal Insulation Mortar | 0.02 | 0.29 | 800 | 1050 | 0.9 | 0.74 |
| | Aerated Concrete Block (B07) | 0.15 | 0.2464 | 700 | 1050 | 0.9 | 0.65 |
| | Thermal Insulation Mortar | 0.02 | 0.29 | 800 | 1050 | 0.9 | 0.74 |
| | Stucco Plaster Pressed into the Mesh Cloth | 0.005 | 0.29 | 800 | 1050 | 0.9 | 0.68 |
| | Lime Cement Mortar | 0.02 | 0.87 | 1700 | 1050 | 0.9 | 0.48 |
| Exterior Wall | Lime Cement Mortar | 0.02 | 0.87 | 1700 | 1050 | 0.9 | 0.48 |
| | Stucco Plaster Pressed into the Mesh Cloth | 0.005 | 0.29 | 800 | 1050 | 0.9 | 0.68 |
| | Thermal Insulation Mortar | 0.02 | 0.29 | 800 | 1050 | 0.9 | 0.74 |
| | Aerated Concrete Block (B07) | 0.25 | 0.33 | 700 | 1050 | 0.9 | 0.65 |
| | Extruded Polystyrene Board | 0.05 | 0.033 | 35 | 1380 | 0.9 | 0.48 |
| | Lime Cement Mortar | 0.025 | 0.87 | 1700 | 1050 | 0.9 | 0.48 |
| Exterior Window | Plate Glass | 0.006 | 0.76 | Solar Transmittance | Solar Reflectance | Visible Transmittance | Visible Reflectance |
| | | | | 0.85 | 0.08 | 0.90 | 0.08 |

**Table 5. Other load settings of office units.**

| Sub-items | Settings |
|---|---|
| Lighting power | 9.0 W/m$^2$ |
| Illuminance standard | 300 lx |
| Number of people per area | 0.25 ppl/m$^2$ |
| New airflow per capita | 30 m$^3$/h |
| Electrical equipment power | 15 W/m$^2$ |
| Infiltration rate per area | 0.0003 m$^3$/s.m$^2$ |
| Ventilation rate per area | 0.0002 m$^3$/s.m$^2$ |
| Min ventilation rate per capita | 0.001 m$^3$/s |
| Recirculated air per area | 0 |
| Ventilation type | no natural ventilation |
| Indoor setpoint temperature | see Fig 5 |

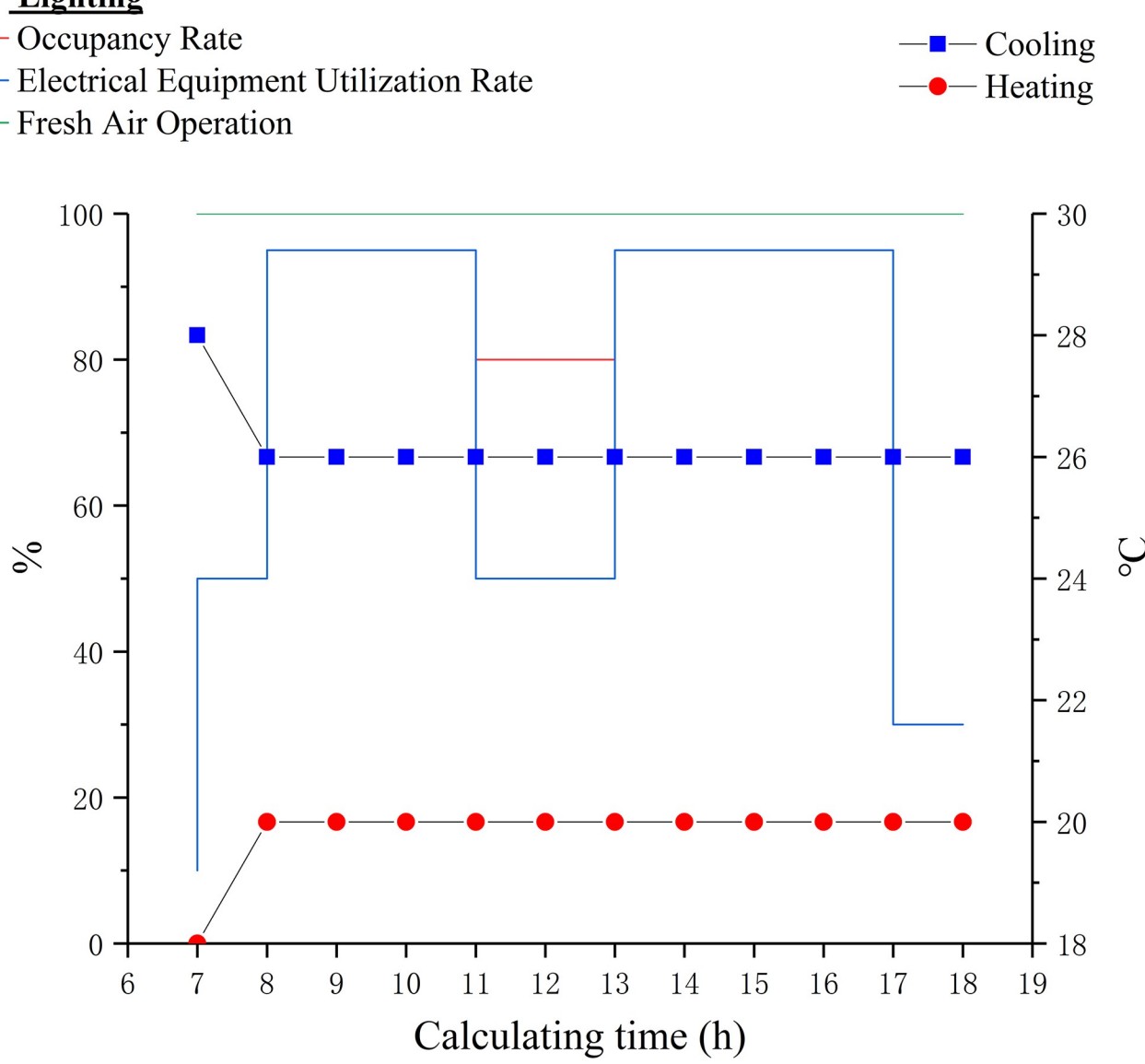

**Fig 5. Building operation schedules.**

## 4 Results and discussion

### 4.1 Results on single-objective optimization

During the single-objective building performance optimization process, the convergence result of the operation tends to be unique and there exists a unique Pareto optimal solution. Fig 6 shows the window opening states when the Pareto solution is reached for different building performance objectives.

From optimization results of each office unit in Fig 6, there are several results: (1) The windowing on the building's north and east facade has a positive effect on reducing AC. Firstly, the north orientation of the building, shaded side, receives the weakest solar radiation compared with other orientations. Secondly, the northeast wind prevails in Yantai summer, and the air convection relieves the indoor heat. West and south facade have strong solar radiation

**Table 6. HVAC system settings.**

| Sub-items | Settings |
|---|---|
| Switch status | always on |
| Heating/Cooling method | hot/cold water coils |
| Fuels | natural gas and electricity |
| Heating/Cooling COP | 3.5 |
| HVAC operation schedule | see Fig 5 |
| Ventilation efficiency | 1 (default) |
| Fan motor efficiency | 0.9 |
| Fan pressure rise | 500 Pa |
| Max heating supply temperature | 40˚C |
| Min cooling supply temperature | 20˚C |
| Rated cooling capacity | 87921.32 W |
| Rated heating capacity | 93782.74 W |

and long sunshine hours, so the windowing area can be minimized, and west facade even does not open windows, which can reduce AC. (2) From the perspective of reducing AH, a lower WWR should be maintained on all building facades, especially on west and north facade, where WWR is as low as 0. Building's east and south facade have a more moderate wind environment, so direct solar heat gain through proper windowing is more beneficial than the others influenced from northwestern cold snap. (3) Under the same conditions, the increase of WWR promotes effectively AL reduction, which is confirmed by arithmetic results in Fig 6, i.e., at the lowest AL, the office space facades all have a high WWR (average WWR > 0.7), while matching to a higher SH or WH to obtain a better solar incidence angle and light area, and then improves indoor daylighting efficiency. (4) Comparing optimal results of AC and AH, this paper also finds that AC and AH show a quantitative relationship of "wane and wax" when optimizing AC or AH, but the sum of the two remains stable.

The optimal results of AC, AH and AL are analyzed above, respectively. In fact, the multi-objective problem can also be transformed into a single-objective problem by conducting a single-objective study on the AT of the office space. The optimization results of AT of each office unit are also recorded in Fig 6, and the feasible solutions to reduce building load are obtained, and then office1 ~ office8 are reorganized to obtain façade windowing schemes of the standard floor. At the lowest AT, by summing the WWR in different façade orientations to take the mean value, and WWR suggested for each orientation of the standard floor is: $WWR_N$ = 0.53, $WWR_W$ = 0.2, $WWR_S$ = 0.6, and $WWR_E$ = 0.43.

## 4.2 Results on multi-objective optimization

Optimal results were presented in Fig 6 on façade windows of each office unit under single-objective optimization. In fact, building energy efficiency research involves synergistic effects between multiple performance objectives, and thus the coupling effects of AC, AH, and AL in each office unit need to be further investigated through a multi-objective optimization approach. In this process, it is generally difficult to find the solution that is optimal for all building performance objectives, i.e., there exists a set of Pareto optimal solutions. The location of the optimal Pareto solution for each building performance objective in the end-generation Pareto set is found by color gradient clustering analysis and marked with A, B, C and D. Fig 7 shows the window opening status of office1 ~ office8 facades under multi-objective coupled optimization under 100 iterations.

| Objectives | Items | Office 1 | Office 2 | Office 3 | Office 4 | Office 5 | Office 6 | Office 7 | Office 8 |
|---|---|---|---|---|---|---|---|---|---|
| AC | Performance |  |  | – | – | – |  |  |  |
|  | AC(Pareto) | **13.516** | **13.250** | **14.395** | **14.349** | **14.338** | **13.958** | **13.831** | **12.994** |
|  | AH | 69.522 | 71.485 | 66.016 | 68.578 | 66.580 | 69.458 | 66.580 | 71.875 |
|  | AL | 15.287 | 12.666 | 25.951 | 25.951 | 25.951 | 16.505 | 18.255 | 11.044 |
|  | AT | 98.324 | 97.01 | 106.361 | 108.878 | 106.868 | 99.921 | 98.665 | 95.913 |
|  | WWR | 0.31 | N=0.41<br>W=0.02 | 0 | W=0<br>S=0 | 0 | S=0<br>E=0.28 | 0.22 | N=0.48<br>E=0.01 |
|  | WH | 1.24 | N=1.80<br>W=2.50 | 0 | W=0<br>S=0 | 0 | S=0<br>E=1.56 | 1.30 | N=1.92<br>E=0.86 |
|  | SH | 2.76 | N=2.20<br>W=1.50 | 0 | W=0<br>S=0 | 0 | S=0<br>E=2.09 | 1.90 | N=2.08<br>E=0.57 |
| AH | Performance | – |  | – |  |  |  |  |  |
|  | AC | 14.201 | 14.405 | 14.395 | 15.400 | 15.971 | 14.136 | 14.492 | 13.899 |
|  | AH(Pareto) | **66.756** | **68.980** | **66.016** | **68.498** | **66.430** | **68.397** | **65.778** | **68.387** |
|  | AL | 25.951 | 25.753 | 25.951 | 25.446 | 23.76 | 23.232 | 25.276 | 24.648 |
|  | AT | 106.907 | 109.137 | 106.361 | 109.345 | 106.161 | 105.765 | 105.545 | 106.934 |
|  | WWR | 0 | N=0.05<br>W=0 | 0 | W=0<br>S=0.11 | 0.18 | S=0<br>E=0.13 | 0.12 | N=0.03<br>E=0.07 |
|  | WH | 0 | N=3.22<br>W=0 | 0 | W=0<br>S=1.50 | 3.90 | S= 0<br>E=3.64 | 1.90 | N=2.80<br>E=3.00 |
|  | SH | 0 | N=0.08<br>W=0 | 0 | W=0<br>S=0.30 | 0.10 | S=0<br>E=0.36 | 0.10 | N=1.20<br>E=1.00 |
| AL | Performance |  |  |  |  |  |  |  |  |
|  | AC | 14.714 | 19.547 | 21.816 | 25.952 | 18.849 | 22.824 | 16.638 | 17.359 |
|  | AH | 72.365 | 77.324 | 69.984 | 73.918 | 67.920 | 72.448 | 68.514 | 76.098 |
|  | AL(Pareto) | **8.858** | **6.305** | **13.259** | **5.770** | **8.083** | **4.919** | **11.116** | **5.536** |
|  | AT | 95.937 | 103.176 | 105.058 | 105.640 | 94.853 | 100.191 | 96.268 | 98.993 |
|  | WWR | 0.94 | N=0.77<br>W=0.76 | 0.93 | W=0.88<br>S=0.86 | 0.79 | S=0.95<br>E=0.74 | 0.79 | N=0.95<br>E=0.81 |
|  | WH | 3.76 | N=3.08<br>W=3.04 | 3.72 | W=3.52<br>S=3.44 | 3.16 | S=3.80<br>E=2.96 | 3.16 | N=3.80<br>E=3.24 |
|  | SH | 0.24 | N=0.92<br>W=0.96 | 0.28 | W=0.48<br>S=0.56 | 0.84 | S=0.20<br>E=1.04 | 0.84 | N=0.20<br>E=0.76 |
| AT | Performance |  |  |  |  |  |  |  |  |
|  | AC | 13.544 | 13.444 | 17.528 | 17.147 | 17.265 | 16.461 | 16.612 | 14.449 |
|  | AH | 70.643 | 72.443 | 68.741 | 69.795 | 67.242 | 70.033 | 68.328 | 72.839 |
|  | AL | 10.028 | 10.028 | 14.863 | 7.948 | 10.864 | 7.502 | 11.678 | 7.528 |
|  | AT(Pareto) | **94.215** | **95.915** | **101.132** | **94.890** | **95.371** | **93.996** | **96.618** | **94.817** |
|  | WWR | 0.61 | N=0.61<br>W=0 | 0.53 | W=0.06<br>S=0.64 | 0.62 | S=0.54<br>E=0.14 | 0.78 | N=0.38<br>E=0.39 |
|  | WH | 2.44 | N=2.44<br>W=0 | 2.12 | W=2.00<br>S=2.56 | 4.00 | S=2.16<br>W=2.93 | 3.87 | N=2.07<br>E=2.21 |
|  | SH | 1.56 | N=1.56<br>W=0 | 1.88 | W=2.00<br>S=1.44 | 0 | S=1.84<br>W=0.43 | 0.10 | N=1.93<br>E=1.79 |

Note: " – " means no windowing on the façade. Unit regarding AC, AH, AL and AT: kWh/m².a; Unit regarding WH and SH: m.

**Fig 6. Optimization results of the facade windowing of office1 ~ office8 under single-objective HypE-GA.**

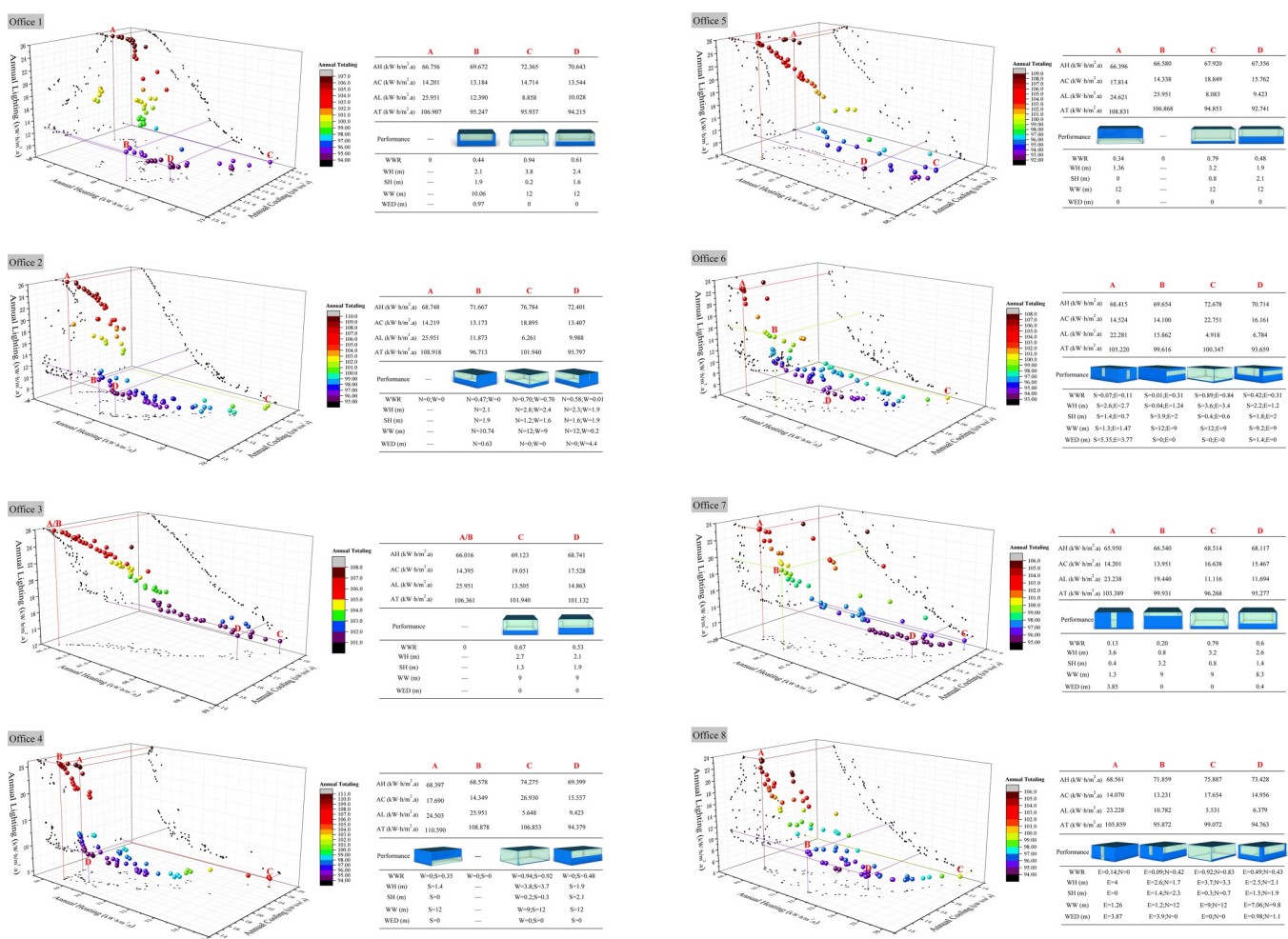

**Fig 7. Building performance simulation results of office1 ~ office8 under 100 iterations.**

It is not easy to obtain optimal solution from Pareto front solutions in the plugin Octopus, so importing genomes and data of all solutions into Origin, redraw 3D scatter plot, and establish chromatographic simple clustering according to the numerical magnitude of AT, which can quickly find the optimal solution from the Pareto set, and analyze the optimal state of individual performance targets. In addition, genomes and target groups corresponding to target points are fed back to the simulation system for testing to confirm the reliability of a design solution.

The four labels, A, B, C and D, are used to denote the Pareto optimal solution of AH, AC, AL, and AT for each office unit, respectively. The operational results verify some findings studied single-objective optimization: (1) AH and AL are negatively correlated in terms of energy consumption quantities as seen from simulation results. The smaller the building windowing in cold region, the less thermal interaction between the interior and exterior, but the higher demand for AL. Conversely, the decrease in AL is due to the increase in WWR, which also leads to the generation of more AH. (2) When AL is the lowest, the building façade in all directions has a high WWR, averaging above 0.7. (3) The south facade has strong solar radiation and long daylight hours, so building AC reaches its lowest point in the state without

windowing. Conversely, building AH is the lowest in the north facade without windowing. (4) The west side of office space has a strong positive correlation between the cold and heat loads, and the negative correlation between each and lighting is also the most significant. windowing design leads to increased heat loss in the room because the U-value of windows is much higher compared to walls, so not windowing or reducing windowing is conducive to generating less building heat and cold loads, but also causes excessive AL. while the windowing on the east side is not significantly affected by outdoor environment and can be opened appropriately to increase building energy consumption.

Label D is the design solution when AT is lowest, considering the balance of energy consumption of AC, AH and AL. Although building heat and cold loads still dominate quantitatively, and fluctuate unsaliently, building AL is more convincing in sensitivity: Calculated by comparison, the AL of middle-side office units on the north, west, south and east directions is 2/5 ~ 3/5 of point A; the AL of corner-side office units on the northwest, southwest, southeast and northeast directions is 1/4 ~ 2/5 of point A. Therefore, the optimal AT in colder regions often depends on lower lighting loads, which means that more effective light area comes from the location of the façade openings. In addition, combined with the expression of office unit façade windowing, it is observed that the windowing at the lowest AT is at the top of façades, i.e., matching the higher SH and WH as much as possible, and the windowing determined based on such geometric logic can optimize the building AT. Based on the Pareto-selected solution of point D, by summing the WWR in different façade orientations to take the mean value, and WWR suggested for each orientation of the standard floor is: $WWR_N = 0.54$, $WWR_W = 0.18$, $WWR_S = 0.46$ and $WWR_E = 0.47$.

## 4.3 Discussion

From the above results of single-objective optimization and multi-objective optimization of building energy consumption, the WWRs-valued of four different windowing orientations are showed very close. In general, designers need to consider buildings' AT more energy efficient, so necessarily refers to the design method of multi-objective optimization. In previous studies, to design more energy efficient windowing solutions for buildings, it is necessary to consider the climate, building types, and constructions and materials of three components, including wall, window frame and glass, but the equally important window geometry factors (WWR, WH, SH, etc.) are seldom considered, or only WWR is highlighted [14]. This study will limit a portion of the building's geometric factors (window directions, bay and depth) and focus on the role of geometric factors (WWR, WH, SH) related to windowing location on the façade on the building's energy efficiency, and it also further improves studies on the windowing geometric factors [16]. This design process is of strong practical significance, both in the early design of façade windowing and in the later reconstruction of façade windowing. Yantai was chosen as the target city for this study, which represents only a very small portion of the cold region, but this research methodology is regionally unrestricted. For designers, the Grasshopper simulation platform in Rhino is more user-friendly, especially surpassing traditional simulation platforms (EnergyPlus, DesignBuilder, DeST, etc.) in terms of human-computer interactivity and diversity of visual representations. In addition, HypE-GA provides more excellent solutions in the process of coupled simulation and optimization of multiple building performances, which share the effort required to be invested in the previous designers' extrapolation and decision-making process compared with the traditional intelligent optimization algorithms [20, 44]. Although HypE-GA improves the computation rate, it still takes a long time. It still consumes a lot of computational time and statistical effort if faced with more complex

projects, i.e., projects involving more design parameters and coupling more building properties. To discuss future research efforts, the findings suggest the following directions:

1. Based on the same methodology, the energy-saving optimization strategies for façade casement windows in different climatic zones and building types in China can be further investigated, so that on the one hand the optimization technology can be promoted, and on the other hand, the performance of local buildings can be improved.

2. For the single façade window opening form of a building, the building's thermal and optical performance (including natural lighting, thermal comfort, window opening behavior, natural ventilation and airtightness, etc.) needs to be further investigated. In addition to this, the optimization of building performance under diverse window opening forms can be further investigated.

3. Under certain conditions, optimization results in the best design strategy. It is still necessary to further consider more objective conditions (thermal properties and construction methods of the facades and windows themselves) and subjective awareness (professional concepts of the designers and needs of the users). In addition, the research system is grounded in the stages of early design and late modification, based on building models-ruled and basic simulation conditions, and its feasibility is yet to be verified in real cases.

## 5 Conclusions

Taking the LB + HB platform and HypE-GA to study on intrinsic relationship between building performances and façade windowing design of standard floor of high-rise office buildings in cold region, and providing support for architects engaged in this aspect. The following summarizes several conclusions of this research.

1. AH and AC are the main components of building energy consumption, but the change of AL has a more significant impact on AT.

2. Single-objective HypE-GA optimization helps to an optimal solution for a designated building performance. To reduce AC, priority is given to windowing on the east and north facade, and it is recommended that WWRN is 0.3 ~ 0.5 and WWRE is 0.2 ~ 0.3; to reduce AH, windows should not be opened on the west and north facade, and the remaining facades should be opened in small areas; to reduce AL, WWR> 0.7 is appropriate for each facade, and matching higher SH or WH should be considered.

3. In terms of building AT, comparing the research results of single-objective HypE-GA optimization and multi-objective HypE-GA optimization, the average WWR of each facade is similar. Multi-objective HypE-GA optimization has more iterations and a higher convergence degree, so the average WWR for each facade of the standard floor is suggested to be $WWR_E = 0.47$, WWRs = 0.46, WWRw = 0.18, and $WWR_N = 0.54$.

4. HypE-GA can be used for rapid iterative search for optimal building performances, effectively reducing the time cost. 3D visualization in Octopus determines the degree of gene population convergence by a color shade, which makes the solution domain of Pareto front solutions not intuitive enough to analyze on the interface, and the 3D scatter simple clustering analysis in Origin can make up for this aspect of Octopus plug-in.

Abbreviations.

| Full Name | Abbreviation | Unit |
|---|---|---|
| Annual Cooling Energy Consumption | AC | kWh/m$^2$.a |
| Annual Heating Energy Consumption | AH | kWh/m$^2$.a |
| Annual Lighting Energy Consumption | AL | kWh/m$^2$.a |
| Annual Totaling Energy Consumption | AT | kWh/m$^2$.a |
| Energy Use Intensity | EUI | kWh/m$^2$.a |
| Hypervolume Estimation Genetic Algorithm | HypE-GA | Unitless |
| Ladybug and Honeybee | LB + HB | Unitless |
| Strength Pareto Evolutionary Algorithm II | SPEA-II | Unitless |
| Nondominated Sorting Genetic Algorithm II | NSGA-II | Unitless |
| Indicator-Based Evolutionary Algorithm | IBEA | Unitless |
| Sill Height | SH | m |
| Window-to-Wall Ratio | WWR | Unitless |
| Window Height | WH | m |
| Window Width | WW | m |
| Window Edge Distance | WED | m |
| North/Northwest/West/Southwest/South/Southeast/East/Northeast | N/NW/W/SW/S/SE/E/NE | Unitless |

## Author Contributions

**Conceptualization:** Weixiang Zhang, Jieli Sui.

**Data curation:** Weixiang Zhang.

**Formal analysis:** Jieli Sui.

**Funding acquisition:** Weixiang Zhang, Jieli Sui.

**Methodology:** Weixiang Zhang, Jieli Sui.

**Resources:** Jieli Sui.

**Software:** Weixiang Zhang.

**Supervision:** Jieli Sui.

**Validation:** Weixiang Zhang, Jieli Sui.

**Visualization:** Weixiang Zhang.

**Writing – original draft:** Weixiang Zhang, Jieli Sui.

**Writing – review & editing:** Weixiang Zhang.

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
