## [Decision Letter · Decision Letter 0]

19 Mar 2024

PONE-D-24-03690HypE-GA Based Optimization Design of Facade Windowing of High-rise Office BuildingsPLOS ONE

Dear Dr. ZHANG WEIXIANG,

Thank you for submitting your manuscript to PLOS ONE. After careful consideration, we feel that it has merit but does not fully meet PLOS ONE’s publication criteria as it currently stands. Therefore, we invite you to submit a revised version of the manuscript that addresses the points raised during the review process.

We look forward to receiving your revised manuscript.

Kind regards,

A A Chowdhury, Ph.D., FHEA, FIEB

Academic Editor

PLOS ONE

Journal Requirements:

"The authors disclosed receipt of the following financial support for the research: the research describe in this paper was funded by Academic Team Project (2022XSTD04) and Youth Foundation Project (2023QNJJ03)."

5. We note that your Data Availability Statement is currently as follows: All relevant data are within the manuscript and its Supporting Information files.

6. "PLOS requires an ORCID iD for the corresponding author in Editorial Manager on papers submitted after December 6th, 2016. Please ensure that you have an ORCID iD and that it is validated in Editorial Manager. To do this, go to ‘Update my Information’ (in the upper left-hand corner of the main menu), and click on the Fetch/Validate link next to the ORCID field. This will take you to the ORCID site and allow you to create a new iD or authenticate a pre-existing iD in Editorial Manager. Please see the following video for instructions on linking an ORCID iD to your Editorial Manager account: https://www.youtube.com/watch?v=_xcclfuvtxQ

Reviewers' comments:

Reviewer's Responses to Questions

**Comments to the Author**

1. Is the manuscript technically sound, and do the data support the conclusions?

Reviewer #1: Partly

Reviewer #2: No

2. Has the statistical analysis been performed appropriately and rigorously? 

Reviewer #1: Yes

Reviewer #2: Yes

3. Have the authors made all data underlying the findings in their manuscript fully available?

Reviewer #1: Yes

Reviewer #2: No

4. Is the manuscript presented in an intelligible fashion and written in standard English?

Reviewer #1: Yes

Reviewer #2: Yes

5. Review Comments to the Author

Reviewer #1: This paper aims to propose a "more complete" optimal design approach oriented towards maximizing energy efficiency, and obtaining the optimal windowing design in high-rise buildings, using HypE-GA in LB+HB platform. The paper is generally interesting for the building performance community.

Some comments:

While the annual cooling, heating, and lighting energy use have been separately observed and analyzed in this case, the overall building performance is also related with and determined by thermal comfort, daylight sufficiency and visual comfort; all of which require a greater multi-objective optimization method. Thus, in that global sense, this paper only offers a more detailed observation on the energy aspect, but not on the whole building performance. Discussion on the impact and relation between energy, thermal, and daylight aspects of the building due to the window design should be provided as well. At the very least, the title must be adjusted to better reflect the limited scope of performance analyzed in this study.

In Eq. (1), what do an1, an2, an3 represent?

Uncommon abbreviations such as AC, AH, AL must be defined first in the text, despite having listed in the List of Abbreviations in p. 1.

In the explanation of Eqs. (1) and (2), the symbol (kW h m^-2 a^-1 ) is just the unit of the performance indicator, which is the Energy Use Intensity (EUI). Thus, the term "min (EUI)" should be employed instead.

In Fig. 3, the length and width of the floorplan should be given in the illustration.

Why are the weather data (TMY) still taken from the year 2005? Any other reasons? How different are the data, compared to more recent weather data?

The Hype-GA assignment settings (Tables 5 and 7) must be written in the Method section, not in the Results. What is the reference for using that settings?

Prior to optimization, there is typically sensitivity analysis conducted to identify the most influential input variables on each output variable (performance indicator). Please consider doing this analysis.

Please provide units for AC, AH, AL in Tables 6 and 8. The term "Pareto" in those tables is misleading, because in a pair of two objectives, there can be many Pareto optimal solutions, instead of only one. What does "Pareto" there actually mean?

What is the computation time for each simulation using HypE-GA in study? How is it compared to simulation with other algorithms?

Discussions are needed in terms of comparison with many relevant studies on the topic of window design optimization, though not necessarily those using GA. What is then the recommended direction for follow-up studies?

Reviewer #2: The paper lacks detailed descriptions of the underlying scientific principles and theoretical frameworks guiding the optimization algorithms used (HypE-GA). Without a clear theoretical foundation, it is challenging for readers to assess the validity and appropriateness of the optimization methods employed.

The technical details of the HypE-GA algorithm, such as convergence mechanisms and mutation strategies, are provided, but there is insufficient explanation or justification for why these specific settings were chosen. Without a clear rationale, readers cannot evaluate the technical accuracy of the optimization process.

The paper lacks context regarding the novelty or significance of the findings. While the optimization results are presented, there is no discussion of how they advance the current understanding of facade windowing design in high-rise office buildings. Without this context, it is difficult to assess the representativeness of the results for the broader scientific community.

Although the optimization settings and results are provided, there is insufficient detail on the simulation methods and input parameters used.

The phrase "This paper investigates..." in the conclusion section represents self-definition. Such phrases should be avoided as they may imply a lack of objectivity and scientific detachment.

The conclusions drawn in the paper lack sufficient empirical support from the data presented. There is a need for more explicit connections between the optimization results and the conclusions drawn regarding the impact of facade windowing on building energy performance.

The paper utilizes artificial intelligence (AI) techniques, specifically the HypE-GA algorithm, for optimization purposes. However, the paper does not clearly specify how AI is integrated into the research methodology, limiting the transparency and reproducibility of the approach.

6. PLOS authors have the option to publish the peer review history of their article (what does this mean?). If published, this will include your full peer review and any attached files.

Reviewer #1: No

Reviewer #2: No

---

## [Author Response · Author response to Decision Letter 0]

31 May 2024

PLoS One 

Dear editor and reviewers,

Thank you for your valuable comments on this study! I have finished revising the research article, PONE-D-24-03690. Original title: HypE-GA Based Optimization Design of Facade Window Opening of High-Rise Office Buildings. Here is my response to the reviewer's comments, a Q&A format.

To Reviewer #1,

Q: While the annual cooling, heating, and lighting energy use have been separately observed and analyzed in this case, the overall building performance is also related with and determined by thermal comfort, daylight sufficiency and visual comfort; all of which require a greater multi-objective optimization method. Thus, in that global sense, this paper only offers a more detailed observation on the energy aspect, but not on the whole building performance. Discussion on the impact and relation between energy, thermal, and daylight aspects of the building due to the window design should be provided as well. At the very least, the title must be adjusted to better reflect the limited scope of performance analyzed in this study.

A: Indeed, the title should focus on specific performance goals, so the title has been adjusted. the new title is “HypE-GA based study on optimal design of standard floor facade windowing of high-rise office buildings facing energy saving in heating, cooling and lighting”. Also, as you said “While the annual cooling, heating, and lighting energy use have been separately observed and analyzed in this case, the overall building performance is also related with and determined by thermal comfort, daylight sufficiency and visual comfort; all of which require a greater multi-objective optimization method.” In our past research, we have also tried to consider coupling more building performance objectives, such as natural daylighting, indoor thermal comfort, and have published our research results in international conferences. Find it at https://doi.org/10.1117/12.2646586

Q: In Eq. (1), what do an1, an2, an3 represent?

A: In order to express the meaning of the formula more clearly, we have made some adjustments to the formula, especially in the letters. an1, an2, an3 are changed in New Eq. (1).

Q: Uncommon abbreviations such as AC, AH, AL must be defined first in the text, despite having listed in the List of Abbreviations in p. 1.

A: All phrases in abbreviated form, whether in the abstract section or in the main text, are given the full name on their first appearance.

Q: In the explanation of Eqs. (1) and (2), the symbol (kW h m^-2 a^-1 ) is just the unit of the performance indicator, which is the Energy Use Intensity (EUI). Thus, the term "min (EUI)" should be employed instead.

A: New Eq (1) and Eq (2), respectively, the symbol (kW h m^-2 a^-1) has been changed.

Q: In Fig. 3, the length and width of the floorplan should be given in the illustration.

A: See new Fig 4 followed. 

Q: Why are the weather data (TMY) still taken from the year 2005? Any other reasons? How different are the data, compared to more recent weather data?

A: Regarding the reasons for choosing the meteorological data for 2005, we have made further additions to the original basis. “The typical meteorological year (2005) data of Yantai Longkou weather station were read by LB. This is a Chinese Standard Weather Data (CSWD) researched and developed by the Meteorological Information Center of the China Meteorological Administration in conjunction with the Department of Building Science of Tsinghua University, which was updated in 2005 and is still commonly used in studies of building simulation. In addition, in 2005, the Yantai region was hit by a huge snowstorm, and the lowest temperature broke the record within 50 years, which was called the year of extremely low temperature, so the meteorological data of that year is particularly important for studying energy conservation in heating demand buildings in northern China.” In addition, it should also be noted that the meteorological data for recent years have been relatively stable compared to 2005, and are therefore not as representative as they could be. For heating buildings in the northern cold zone, it is necessary to consider the historical extreme cold years as the meteorological conditions for simulation.

Q: The Hype-GA assignment settings (Tables 5 and 7) must be written in the Method section, not in the Results. What is the reference for using that settings?

A: Tables 5 and 7 have been replaced by new Tables 1 and 2, which located in the Optimization algorithm subsection of the Methodology section. The setting of the algorithm parameters in the table is also explained. “The operating parameters, including Elitism, Mutation Rate, Mut.Probability, Crossover Rate, Population Size and Max Generations, are very critical because different settings of parameters will have a large impact on the efficiency of HypE-GA. In the optimal process, Elitism gives at least 50% new solutions that are bred out of the Elite instead of the entire pool. The lower Crossover Rate, the less information is exchanged between individuals, resulting in undercharacterised new solutions, so a value of 0.4 ~ 0.99 is recommended. A low Mutation Rate means little changes to the parameters’ values, multiplied by Mut.Probability is generally recommended to take a value of 0.0001 ~ 0.1. Population Size and Max Generations directly affects the efficiency of HypE-GA, which can be set a value according to the case complexity in 10 ~ 50 and in 10 ~ 200, respectively. In this case study, based on the above principles, operating parameters of single-objective HypE-GA are shown in Table 1, and that of multi-objective HypE-GA are shown in Table 2.”

Q: Prior to optimization, there is typically sensitivity analysis conducted to identify the most influential input variables on each output variable (performance indicator). Please consider doing this analysis.

A: In our previous study, we have done the sensitivity analysis of WWR,WH,SH with AH,AC,AL and AT as the building performance targets, and get some conclusions like different factors have varied influences on AT under different orientations: WWRS > WWRN > WWRE > WWRW; WHE > WHS > WHN > WHW; SHS > SHE > SHN > SHW. Finding it at https://link.cnki.net/doi/10.13951/j.cnki.37-1213/n.210112.

Q: Please provide units for AC, AH, AL in Tables 6 and 8. The term "Pareto" in those tables is misleading, because in a pair of two objectives, there can be many Pareto optimal solutions, instead of only one. What does "Pareto" there actually mean?

A: Units are displayed in the notes below the Tables 6, which has been changed as new Tables 7. In order to better express the correspondence between the data in Table 8 and the scatterplot, we have merged them into what is new Figure 6, and the units are labeled in the figure .

Regarding the interpretation of pareto, we apologize for not explaining it in our previous article, which led to a misunderstanding on the part of our readers. Now, we make the explanation in single-objective optimization simulation and multi-objective optimization simulation respectively, so that the reader understands the meaning of pareto in single/objective optimization results. We have made additions in Results and discussion section, “During the single-objective building performance optimization process, the convergence result of the operation tends to be unique and there exists a unique Pareto optimal solution. Table 7 shows the window opening states when the Pareto solution is reached for different building performance objectives.” and “building energy efficiency research involves synergistic effects between multiple performance objectives, and thus the coupling effects of AC, AH, and AL in each office unit need to be further investigated through a multi-objective optimization approach. In this process, it is generally difficult to find the solution that is optimal for all building performance objectives, i.e., there exists a Pareto set solutions. The location of the optimal Pareto solution for each building performance objective in the end-generation Pareto solution set is found by color gradient clustering analysis and marked with A, B, C and D. Fig. 5 shows the window opening status of office1 ~ office8 facades under multi-objective coupled optimization under 100 iterations. The four labels, A, B, C and D, are used to denote the Pareto optimal solution of AH, AC, AL, and AT for each office unit, respectively.”

Q: What is the computation time for each simulation using HypE-GA in study? How is it compared to simulation with other algorithms?

A: Regarding computation time for each simulation using HypE-GA in study, I'm sorry we haven't kept statistical records. From memory we calculate it in about 2 minutes a time, but it takes longer to iterate it once. In single-objective optimization, 20 iterations total about 260 hours; in multi-objective optimization, 100 iterations total about 730 hours.The fact that we treat each office unit as a separate run file, and therefore cannot run in parallel, adds significantly to our time costs. Single- and multi-objective run files we will submit with the article to ensure the operationalization and generalization of our design process.

HypE-GA has unique advantages over the three evolutionary algorithms with which we are most familiar. We add in the paper “HypE-GA uses Monte Carlo simulations to approximate the exact hypervolume values. The main idea is that the actual metric value is not important, but the ranking of solutions induced by the hypervolume metric is important. Thus, hypervolume-based search not only solves multi-objective problems, but also flexibly adjusts the running time. Bader and Zitzler showed HypE-GA was used for more accuracy and speed than similar algorithms, including Strength Pareto Evolutionary Algorithm Ⅱ (SPEA-Ⅱ) , Nondominated Sorting Genetic Algorithm Ⅱ (NSGA-Ⅱ) and Indicator-Based Evolutionary Algorithm (IBEA), in the optimization process in cases with more than three fitness functions. The HypE-GA was matched with HypE Reduction convergence mechanism and HypE Mutation mechanism. Hype Reduction stipulated how to truncate a Pareto front solutions to fit the result size when it is too big. HypE Mutation applied mutation and recombination operators to generate N offspring to simulate the diversity of iterative processes. Therefore, HypE-GA is an appropriate way to solve single-objective or multi-objective optimization problems in this study.” Find it at https://www.sci-hub.yt/10.1162/evco_a_00009

Q: Discussions are needed in terms of comparison with many relevant studies on the topic of window design optimization, though not necessarily those using GA. What is then the recommended direction for follow-up studies?

To discuss future work, the findings provide the following recommendations and implications.

(1) Based on the same methodology, the energy-saving optimization strategies for façade casement windows in different climatic zones and building types in China can be further investigated, so that on the one hand the optimization technology can be promoted, and on the other hand, the performance of local buildings can be improved. 

(2) For the single façade window opening form of a building, the building's thermal and optical performance (including natural lighting, thermal comfort, window opening behavior, natural ventilation and airtightness, etc.) needs to be further investigated. In addition to this, the optimization of building performance under diverse window opening forms can be further investigated. 

(3) Under certain conditions, optimization results in the best design strategy. It is still necessary to further consider more objective conditions (thermal properties and construction methods of the facades and windows themselves) and subjective awareness (professional concepts of the designers and needs of the users). In addition, the research system is grounded in the stages of early design and late modification, based on building models-ruled and basic simulation conditions, and its feasibility is yet to be verified in real cases.

To Reviewer #2,

Q: The paper lacks detailed descriptions of the underlying scientific principles and theoretical frameworks guiding the optimization algorithms used (HypE-GA). Without a clear theoretical foundation, it is challenging for readers to assess the validity and appropriateness of the optimization methods employed.

A: HypE-GA is one of the GA's that are characterized by advantages in computational processes with many nonlinearities formed by high-dimensional design parameters and three and more optimization objectives. Firstly, in the process of many iterative calculations, there will not be a bureau early convergence Pareto solution set, but the generation is more excellent to develop. Secondly, the use of Monte Carlo simulation to approximate the exact hypervolume values improves the efficiency and accuracy of model data optimization. Third, the ranking of monotonicity excellent solutions is always maintained in the results of each generation of optimization, which also saves the time and effort of manual data processing.

Q: The technical details of the HypE-GA algorithm, such as convergence mechanisms and mutation strategies, are provided, but there is insufficient explanation or justification for why these specific settings were chosen. Without a clear rationale, readers cannot evaluate the technical accuracy of the optimization process.

A: The paper adds the connotation of HypE Reduction convergence mechanism and HypE Mutation mechanism. i.e., “The HypE-GA was matched with HypE Reduction convergence mechanism and HypE Mutation mechanism. The set of objective value vectors corresponding to each solution in the Pareto Set is called the Pareto Front. Hype Reduction stipulated how to truncate a Pareto front solutions to fit the result size when it is too big. HypE Mutation applied mutation and recombination operators to generate N offspring to simulate the diversity of iterative processes.” And “A low Mutation Rate means little changes to the parameters’ values, multiplied by Mut.Probability is generally recommended to take a value of 0.0001 ~ 0.1.” Because of the possibility of mutations in the N new individuals produced by crossover, this is more in line with the laws of nature, but in order to ensure a smooth iteration, generally go to a very small value, 0.0001~0.1.

Q: The paper lacks context regarding the novelty or significance of the findings. While the optimization results are presented, there is no discussion of how they advance the current understanding of facade windowing design in high-rise office buildings. Without this context, it is difficult to assess the representativeness of the results for the broader scientific community.

A: At the end of this part of the introduction, we add a description of the innovative and contextual nature of this study, explaining that this type of research is topical at the moment and that the innovations focus on the design methodology. “Nowadays, the green concept in green building, urban renewal and remodeling has attracted much attention.In previous architectural research and practice, there are numerous software for simulating building energy efficiency, but their single functionality and weak interaction between software cannot fully meet the design needs of architects. Therefore, on the one hand, to solve the problems of unfriendly multi-platform interactivity and low optimization efficiency, especially for design projects based on multi-objective optimization. On the other hand, to provide architects with objective and fast decision support for early design and later remodeling of façade windowing. Studying the relationship between windowing design and single/multi-objective building performances in high-rise office buildings in cold region applying through LB + HB and Hypervolume Estimation Genetic Algorithm (HypE-GA), which becomes an innovative aspect that differs from above studies.”

The novelty and accuracy of the results are expected to be further verified through the practice of engineering projects. Not only will the results of this study be verified, but also more design parameters and more performance objectives will be incorporated to optimise the indoor environment of the 

---

## [Decision Letter · Decision Letter 1]

16 Jun 2024

PONE-D-24-03690R1HypE-GA based study on optimal design of standard floor facade windowing of high-rise office buildings facing energy saving in heating, cooling and lightingPLOS ONE

Dear Dr. ZHANG WEIXIANG,

Thank you for submitting your manuscript to PLOS ONE. After careful consideration, we feel that it has merit but does not fully meet PLOS ONE’s publication criteria as it currently stands. Therefore, we invite you to submit a revised version of the manuscript that addresses the points raised during the review process.

Below are the comments of a reviewer regarding your submission to PLOS One. The  reviewer has made substantive critical comments, and you should pay close attention to them when making your revisions. The reviewer comments are important, as they will assist you in making your paper much more interesting to our readers. Please, address properly all the comments of the reviewer. Please, submit your revised manuscript online by using the Editorial Manager system.

We look forward to receiving your revised manuscript.

Kind regards,

Ashfaque Ahmed Chowdhury, Ph.D., FHEA, FIEB

Academic Editor

PLOS ONE

**Comments to the Author**

1. If the authors have adequately addressed your comments raised in a previous round of review and you feel that this manuscript is now acceptable for publication, you may indicate that here to bypass the “Comments to the Author” section, enter your conflict of interest statement in the “Confidential to Editor” section, and submit your "Accept" recommendation.

Reviewer #1: (No Response)

Reviewer #2: (No Response)

2. Is the manuscript technically sound, and do the data support the conclusions?

Reviewer #1: Yes

Reviewer #2: Partly

3. Has the statistical analysis been performed appropriately and rigorously? 

Reviewer #1: Yes

Reviewer #2: N/A

4. Have the authors made all data underlying the findings in their manuscript fully available?

Reviewer #1: Yes

Reviewer #2: No

5. Is the manuscript presented in an intelligible fashion and written in standard English?

Reviewer #1: No

Reviewer #2: Yes

6. Review Comments to the Author

Reviewer #1: The manuscript has been revised from the original version.

While most of the reviewers' comments have been addressed, there are still some issues left.

The opening sentence in the Abstract ("Taking the LB + HB platform...") is a bit difficult to understand and is not commonly found as a typical abstract opening sentence. Perhaps the wordings should be rephrased. A general background shall be provided, followed with the knowledge gap and aims or objectives of the study.

*All* responses to the reviewers' comments should be included in the revised manuscript, in the appropriate places; not only in the response/rebuttal document. Please ensure this has been incorporated.

For example, in some of their responses, the authors refer to their previous works, but those works are not mentioned or properly cited in the text. We should not assume that the readers would have been familiar with those previous works, so the current manuscript should concisely describe and cite them.

Definitions of Window Width (WW) and Window Edge Distance (WED) should be provided with illustrations.

In the multi-objective optimization, in a given pair of two conflicting objectives, there can be many Pareto optimal solutions (in plural, with 's'). These optimal solutions altogether are called the Pareto optimal set, or just Pareto set, or Pareto front (in singular). Thus, be careful and consistent when referring to those terminologies. For instance, in line 306: "there exists a Pareto set solutions" is not appropriate. It should be "there exists a Pareto optimal set", or "there exists a set of Pareto optimal solutions" (although the former is preferrable).

In lines 311 and 314: "Pareto front solutions" should be "the Pareto set". Please check for any similar occurrences.

A dedicated Discussion or General Discussion section is recommended, prior to the Conclusion section. See comment from the previous review regarding comparison with many relevant studies on this topic, and recommended direction for follow-up studies. These all, together with the limitations of the methods, shall be placed in a separate Discussion section. Moreover, responses to the reviewers' comments that are not directly related to the main methods and findings of this study can also be placed in that Discussion section as well.

Reviewer #2: The paper focuses on optimizing the design of facade windowing in high-rise office buildings in Yantai, a cold region, using the Ladybug and Honeybee (LB + HB) platform and the Hypervolume Estimation Genetic Algorithm (HypE-GA). The primary objectives are to reduce annual cooling (AC), heating (AH), and lighting (AL) energy consumption by manipulating the window-to-wall ratio (WWR), window height (WH), and sill height (SH). The research is valuable for practical applications and contributes incrementally to the existing body of knowledge. However, it does not offer substantial scientific advancements or novel theories that significantly push the boundaries of the field. To enhance its scientific value, the study should incorporate a more comprehensive performance analysis and address the broader implications of the findings.

The study focuses predominantly on energy consumption (AC, AH, AL) without adequately integrating other crucial aspects of building performance such as thermal comfort, daylight sufficiency, and visual comfort. This limitation reduces the holistic applicability of the findings in real-world scenarios where occupant comfort is equally important.

the absence of statistical validation and sensitivity analysis weakens the robustness of the findings from a scientific standpoint.

The study uses Typical Meteorological Year (TMY) data from 2005 without justifying why this dataset was chosen over more recent data. Climate data can significantly affect energy performance simulations, and using outdated data might not reflect current or future conditions accurately.

Some methodological details are vague, such as the explanation of key parameters in equations and the rationale behind specific optimization choices.

Uncommon abbreviations such as AC, AH, and AL should be defined clearly when first mentioned in the text.

Figures such as floor plans should include dimensions for better understanding. Additionally, the clarity and labeling of charts and graphs could be improved to enhance interpretability.

7. PLOS authors have the option to publish the peer review history of their article (what does this mean?). If published, this will include your full peer review and any attached files.

Reviewer #1: No

Reviewer #2: **Yes: **Esmatullah Noorzai, Lecturer, Department of Project and Construction Management, University of Tehran, Tehran, 1415564583, Iran. ORCID:

---

## [Author Response · Author response to Decision Letter 1]

29 Jun 2024

PLoS One 

Dear editor and reviewers,

Thank you for your valuable comments on this study! I have finished revising the research article, PONE-D-24-03690. Paper title: HypE-GA based study on optimal design of standard floor facade windowing of high-rise office buildings facing energy saving in heating, cooling and lighting. Please pay special attention to the following additions to my grant fund: The authors disclosed receipt of the following financial support for the research, authorship and publication of this article: The research described in this paper was funded by Yantai Institute of Technology Youth Foundation Program (2023QNJJ03), Yantai Institute of Technology Academic Team Program (2022XSTD04) and Shandong Social Science Planning Research Program (22CWYJ35). The funders had no role in study design, data collection and analysis, decision to publish, or preparation of the manuscript.

In addition, here is my response to the reviewer's comments, a Q&A format. Sections in red indicate additions or modifications as the below.

To Reviewer #1,

Q: The opening sentence in the Abstract ("Taking the LB + HB platform...") is a bit difficult to understand and is not commonly found as a typical abstract opening sentence. Perhaps the wordings should be rephrased. A general background shall be provided, followed with the knowledge gap and aims or objectives of the study.

A: The quantitative design on area and location of building façade’s windows has a significant impact on interior light and heat environment, which is also very instructive for preliminary and remodeling design of buildings. However, previous studies paid more attention to the thermal insulation construction and shading based on design parameters from the perspective of designers, but neglected the fact that the geometric properties of the windows themselves are equally important for building energy efficiency. Secondly, the weak interactivity and algorithmic limitations of traditional simulation platforms prevent rapid access to ideal design strategies. Therefore, this paper takes the standard floor of a high-rise office building as the research object in cold region－Yantai, facing façade windowing design, the three building performance objectives of each office unit－Annual Cooling Energy Consumption (AC), Annual Heating Energy Consumption (AH) and Annual Lighting Energy Consumption (AL)－are simulated and single/multi-objective optimized by relying on Ladybug and Honeybee (LB + HB) platform and Hypervolume Estimation Genetic Algorithm (HypE-GA) to obtain the genome of Pareto－Window-to-Wall Ratio (WWR), Window Height (WH) and Sill Height (SH)－at the lowest of each performance objective in order to determine the most energy-efficient façade windowing expression. The results show that AH and AC, their sum of quantities remains stable, are main energy consumption sources of office buildings, while the change of AL is more likely to have an impact than the others’ on Annual Totaling Energy Consumption (AT). The analysis points out that different windowing strategies can be adopted for different performance objectives. To reduce AC, priority is given to windowing on the east and north facade, with East Window-to-Wall Ratio (WWRE) at 0.2 ~ 0.3 and North Window-to-Wall Ratio (WWRN) at 0.3 ~ 0.5; to reduce AH, windows on the west and north facade should not be opened, and the remaining facades should be opened in small areas; to reduce AL, WWR> 0.7 is appropriate for each facade, and should be considered to matching a higher SH or WH; From AT, the average WWR in the single-objective and multi-objective optimization results are similar, so it is suggested that the WWR of each facade of office buildings in Yantai area is WWRE=0.47, North South Window-to-Wall Ratio (WWRS)=0.46, West Window-to-Wall Ratio (WWRW)=0.18 and WWRN=0.54. In addition, this paper proposes a method that can quickly find the Pareto optimal solution by clustering analysis on optimized results through Origin in multi-objective HypE-GA optimization study.

Q: in some of their responses, the authors refer to their previous works, but those works are not mentioned or properly cited in the text. We should not assume that the readers would have been familiar with those previous works, so the current manuscript should concisely describe and cite them.

A: In the authors' previous studies, two parts of the work were specifically done. Not only verified that window geometric parameters have the greatest sensitivity to building performance, but also remedied the above two research shortcomings. In the first part, the multicollinearity problem between high-dimensional design parameters and multi-objective building performance (building energy consumption, daylighting, indoor thermal comfort) was solved by principal component analysis. It was found that only 12 of the preset 15 parameters passed the KMO Sampling Suitability Quantity and BARTLETT sphericity test, and they were classified into four principal components: Building Body (Bay, Depth and WWR), Windowing Geometry (Window Height, Sill Height and Window Horizontal Distance), Window Properties (K-value, Solar Heat Gain Coefficient and Visual Transmittance), and Facade Properties (Wall Height, Plane Corner, External Wall R-Value), and the ensuing weight calculations yielded that the weights of each principal component were 0.41, 0.268, 0.188, and 0.134 in that order, thus proposing the principal component linear function assessment model. In the design of buildings with Bay and Depth, the Windowing Geometry with WWR is the most sensitive type of design parameters for building performance [24]. In the second part, the sensitivity of window geometric parameters to building energy consumption (AC, AH, AL and AT) was investigated by variance calculation. It is shown that WWR is the dominant factor affecting indoor AC and AH in the design of facade windowing, and larger values of WWR, WH and SH within a certain range are preferable in terms of AT reduction. The best orientation for window-opening is south, east, north, and west in a sequence when WWR is in the range of 0.3 to 0.7. Different factors have varied influences on AT under different orientations: WWRS > WWRN > WWRE > WWRW；WHE > WHS > WHN > WHW；SHS > SHE > SHN > SHW[25]。

These are the research methodology and some of the conclusions from previous studies, in this paper in references [24], [25]. In addition, there are images of the simulation program mentioned in the last reply letter that have been shared as modelling details in a zip package, all uploaded to the system.

Q: Definitions of Window Width (WW) and Window Edge Distance (WED) should be provided with illustrations.

A: Additional information in the paper “Horizontally, the WW is the width of the window opening and the WED is the distance from the side of the window opening to the demarcation line of the office unit.” see Fig 4.

Q: In the multi-objective optimization, in a given pair of two conflicting objectives, there can be many Pareto optimal solutions (in plural, with 's'). These optimal solutions altogether are called the Pareto optimal set, or just Pareto set, or Pareto front (in singular). Thus, be careful and consistent when referring to those terminologies. For instance, in line 306: "there exists a Pareto set solutions" is not appropriate. It should be "there exists a Pareto optimal set", or "there exists a set of Pareto optimal solutions" (although the former is preferrable).In lines 311 and 314: "Pareto front solutions" should be "the Pareto set". Please check for any similar occurrences.

A: Forms of words have been modified in the paper. “Optimal results were presented in Table7 on façade windows of each office unit under single-objective optimization. In fact, building energy efficiency research involves synergistic effects between multiple performance objectives, and thus the coupling effects of AC, AH, and AL in each office unit need to be further investigated through a multi-objective optimization approach. In this process, it is generally difficult to find the solution that is optimal for all building performance objectives, i.e., there exists a set of Pareto optimal solutions. The location of the optimal Pareto solution for each building performance objective in the end-generation Pareto set is found by color gradient clustering analysis and marked with A, B, C and D. Fig. 6 shows the window opening status of office1 ~ office8 facades under multi-objective coupled optimization under 100 iterations.

It is not easy to obtain optimal solution from Pareto front solutions in the plugin Octopus, so importing genomes and data of all solutions into Origin, redraw 3D scatter plot, and establish chromatographic simple clustering according to the numerical magnitude of AT, which can quickly find the optimal solution from the Pareto set, and analyze the optimal state of individual performance targets. In addition, genomes and target groups corresponding to target points are fed back to the simulation system for testing to confirm the reliability of a design solution.”

Q: A dedicated Discussion or General Discussion section is recommended, prior to the Conclusion section. See comment from the previous review regarding comparison with many relevant studies on this topic, and recommended direction for follow-up studies.

A: A Discussion subsection was added before the Conclusion section, where the progress made in the research results over the previous research articles was discussed, as well as the problems that still exist. Even future research directions are discussed. “From the above results of single-objective optimization and multi-objective optimization of building energy consumption, the WWRs-valued of four different windowing orientations are showed very close. In general, designers need to consider buildings’ AT more energy efficient, so necessarily refers to the design method of multi-objective optimization. In previous studies, to design more energy efficient windowing solutions for buildings, it is necessary to consider the climate, building types, and constructions and materials of three components, including wall, window frame and glass, but the equally important window geometry factors (WWR, WH, SH, etc.) are seldom considered, or only WWR is highlighted [14]. This study will limit a portion of the building's geometric factors (window directions, bay and depth) and focus on the role of geometric factors (WWR, WH, SH) related to windowing location on the façade on the building's energy efficiency, and it also further improves studies on the windowing geometric factors [16]. This design process is of strong practical significance, both in the early design of façade windowing and in the later reconstruction of façade windowing. Yantai was chosen as the target city for this study, which represents only a very small portion of the cold region, but this research methodology is regionally unrestricted. For designers, the Grasshopper simulation platform in Rhino is more user-friendly, especially surpassing traditional simulation platforms (EnergyPlus, DesignBuilder, DeST, etc.) in terms of human-computer interactivity and diversity of visual representations. In addition, HypE-GA provides more excellent solutions in the process of coupled simulation and optimization of multiple building performances, which share the effort required to be invested in the previous designers' extrapolation and decision-making process compared with the traditional intelligent optimization algorithms [20,44]. Although HypE-GA improves the computation rate, it still takes a long time. It still consumes a lot of computational time and statistical effort if faced with more complex projects, i.e., projects involving more design parameters and coupling more building properties. To discuss future research efforts, the findings suggest the following directions:

 (1) Based on the same methodology, the energy-saving optimization strategies for façade casement windows in different climatic zones and building types in China can be further investigated, so that on the one hand the optimization technology can be promoted, and on the other hand, the performance of local buildings can be improved. 

(2) For the single façade window opening form of a building, the building's thermal and optical performance (including natural lighting, thermal comfort, window opening behavior, natural ventilation and airtightness, etc.) needs to be further investigated. In addition to this, the optimization of building performance under diverse window opening forms can be further investigated. 

(3) Under certain conditions, optimization results in the best design strategy. It is still necessary to further consider more objective conditions (thermal properties and construction methods of the facades and windows themselves) and subjective awareness (professional concepts of the designers and needs of the users). In addition, the research system is grounded in the stages of early design and late modification, based on building models-ruled and basic simulation conditions, and its feasibility is yet to be verified in real cases.”

To Reviewer #2,

Q: The study focuses predominantly on energy consumption (AC, AH, AL) without adequately integrating other crucial aspects of building performance such as thermal comfort, daylight sufficiency, and visual comfort. This limitation reduces the holistic applicability of the findings in real-world scenarios where occupant comfort is equally important. the absence of statistical validation and sensitivity analysis weakens the robustness of the findings from a scientific standpoint.

A: The focus of this article's research is to observe the effect of window geometric features on AC, AL, and AH through their design parameters and to obtain an energy efficient window opening strategy. In the authors' previous studies, the intrinsic link between window opening (considering the physical properties of the adjacent envelope) on lighting, energy consumption, and thermal comfort in building interiors has been investigated on the one hand, as detailed in Ref. [24].On the other hand, the sensitivity of the design parameters(WWR, AH, SH) to AC, AL, and AH has been studied, and the results of their research are mentioned in the introductory section of this article, as detailed in Ref. [25]. In the Introduction Section of this article, a review of the important conclusions of the authors' previous studies in these two aspects.

Q: The study uses Typical Meteorological Year (TMY) data from 2005 without justifying why this dataset was chosen over more recent data. Climate data can significantly affect energy performance simulations, and using outdated data might not reflect current or future conditions accurately.

A: Regarding the reasons for choosing the meteorological data for 2005, we have made further additions to the original basis. “The typical meteorological year (2005) data of Yantai Longkou weather station were read by LB. This is a Chinese Standard Weather Data (CSWD) researched and developed by the Meteorological Information Center of the China Meteorological Administration in conjunction with the Department of Building Science of Tsinghua University, which was updated in 2005 and is still commonly used in studies of building simulation. In addition, in 2005, the Yantai region was hit by a huge snowstorm, and the lowest temperature broke the record within 50 years, which was called the year of extremely low temperature, so the meteorological data of that year is particularly important for studying energy conservation in heating demand buildings in northern China.” In addition, it should also be noted that the meteorological data for recent years (2006 to 2022) have been relatively stable compared to 2005, and are therefore not as representative as they could be. For heating buildings in the current northern cold zone, it is necessary to consider the historical extreme cold years as the meteorological conditions for simulation.

Q: Some methodological details are vague, such as the explanation of key parameters in equations and the rationale behind specific optimization choices.

A: The key parameters and optimization principles in the formulation have been further supplemented in the pape

---

## [Decision Letter · Decision Letter 2]

8 Aug 2024

PONE-D-24-03690R2HypE-GA based study on optimal design of standard floor facade windowing of high-rise office buildings facing energy saving in heating, cooling and lightingPLOS ONE

Dear Dr. ZHANG WEIXIANG,

Thank you for submitting your manuscript to PLOS ONE. After careful consideration, we feel that it has merit but does not fully meet PLOS ONE’s publication criteria as it currently stands. Therefore, we invite you to submit a revised version of the manuscript that addresses the points raised during the review process. Please submit your revised manuscript by Sep 22 2024 11:59PM. If you will need more time than this to complete your revisions, please reply to this message or contact the journal office at plosone@plos.org. Please include the following items when submitting your revised manuscript:A rebuttal letter that responds to each point raised by the academic editor and reviewer(s). You should upload this letter as a separate file labeled 'Response to Reviewers'.A marked-up copy of your manuscript that highlights changes made to the original version. You should upload this as a separate file labeled 'Revised Manuscript with Track Changes'.An unmarked version of your revised paper without tracked changes. You should upload this as a separate file labeled 'Manuscript'.If applicable, we recommend that you deposit your laboratory protocols in protocols.io to enhance the reproducibility of your results. Protocols.io assigns your protocol its own identifier (DOI) so that it can be cited independently in the future. For instructions see: https://journals.plos.org/plosone/s/submission-guidelines#loc-laboratory-protocols. Additionally, PLOS ONE offers an option for publishing peer-reviewed Lab Protocol articles, which describe protocols hosted on protocols.io. Read more information on sharing protocols at https://plos.org/protocols?utm_medium=editorial-email&utm_source=authorletters&utm_campaign=protocols.

We look forward to receiving your revised manuscript.

Kind regards,

Ashfaque Ahmed Chowdhury, Ph.D., FHEA, FIEB

Academic Editor

PLOS ONE

Journal Requirements:

Reviewers' comments:

Reviewer's Responses to Questions

**Comments to the Author**

1. If the authors have adequately addressed your comments raised in a previous round of review and you feel that this manuscript is now acceptable for publication, you may indicate that here to bypass the “Comments to the Author” section, enter your conflict of interest statement in the “Confidential to Editor” section, and submit your "Accept" recommendation.

Reviewer #1: All comments have been addressed

Reviewer #2: (No Response)

2. Is the manuscript technically sound, and do the data support the conclusions?

Reviewer #1: Yes

Reviewer #2: Partly

3. Has the statistical analysis been performed appropriately and rigorously? 

Reviewer #1: Yes

Reviewer #2: Yes

4. Have the authors made all data underlying the findings in their manuscript fully available?

Reviewer #1: Yes

Reviewer #2: (No Response)

5. Is the manuscript presented in an intelligible fashion and written in standard English?

Reviewer #1: Yes

Reviewer #2: (No Response)

6. Review Comments to the Author

**Reviewer #1: **The manuscript has been revised and would be almost ready for publication.

Some small comments:

In p 3, lines 77-78, the sentence "In the authors' previous studies, ..." can be set as the beginning of a new paragraph, to avoid having a very long paragraph. It is perhaps also more elegant to not directly referring the works as the authors'; so for instance: "In two earlier studies / works [24,25], two parts of the work were specifically done."

**Reviewer #2:** he study focuses on optimizing window design for energy efficiency in high-rise office buildings using a combination of simulation platforms and genetic algorithms. While the topic is relevant, the research lacks a comprehensive theoretical framework that ties the findings to broader principles in building science or energy efficiency. The paper is more applied in nature, which raises questions about its contribution to scientific knowledge beyond the specific case study of Yantai

The paper claims to present a new approach to optimizing window design, but it does not clearly demonstrate how this approach represents a significant scientific advance. The study appears to be a practical application of existing tools rather than a novel contribution to the field.

The conclusions are presented without adequate support from the data. The paper makes broad claims about the effectiveness of the proposed window design strategies without fully demonstrating how the data supports these claims.

The paper does not demonstrate an adequate understanding of the relevant literature. While it references some studies, it fails to engage deeply with significant works that could provide context or challenge the findings. Important literature on building energy efficiency and genetic algorithms is either overlooked or inadequately discussed

The conclusions drawn from the results are not fully supported by the data, and there is a disconnect between the results and the broader implications of the study. The paper fails to adequately tie together its findings with the existing body of knowledge.

The paper does not clearly identify implications for research, practice, or society. It fails to bridge the gap between theory and practice, and the practical applications of the research are not well articulated. The societal impact of the research is not discussed, limiting its relevance and value.

7. PLOS authors have the option to publish the peer review history of their article (what does this mean?). If published, this will include your full peer review and any attached files.

Reviewer #1: No

Reviewer #2: **Yes: **Esmatullah Noorzai Lecturer, Dept. of Project and Construction Management, School of Architecture, Univ. of Tehran, Tehran

---

## [Author Response · Author response to Decision Letter 2]

9 Aug 2024

PLoS One 

Dear editor and reviewers,

Thank you for your valuable comments on this study! I have finished revising the research article, PONE-D-24-03690. Paper title: HypE-GA based study on optimal design of standard floor facade windowing of high-rise office buildings facing energy saving in heating, cooling and lighting. Please pay special attention to the following additions to my grant fund: The authors disclosed receipt of the following financial support for the research, authorship and publication of this article: The research described in this paper was funded by Yantai Institute of Technology Youth Foundation Program (2023QNJJ03), Yantai Institute of Technology Academic Team Program (2022XSTD04) and Shandong Social Science Planning Research Program (22CWYJ35). The funders had no role in study design, data collection and analysis, decision to publish, or preparation of the manuscript.

In addition, here is my response to the reviewer's comments, a Q&A format. Sections in red indicate additions or modifications as the below.

To Reviewer #1,

Q: In p 3, lines 77-78, the sentence "In the authors' previous studies, ..." can be set as the beginning of a new paragraph, to avoid having a very long paragraph. It is perhaps also more elegant to not directly referring the works as the authors'; so for instance: "In two earlier studies / works [24,25], two parts of the work were specifically done."

A: In the authors' previous studies, scholars often selected solutions by performing sensitivity analysis on design parameters [21,22,23], but two drawbacks are obvious: (1) the sensitive weights of design parameters are not assigned precisely enough; (2) the base group of design solutions expands exponentially as the dual increase of design parameters’ number and performance indicators’ dimension, thus consuming a lot of time and effort. In two earlier studies [24,25], two parts of the work were specifically done.

To Reviewer #2,

Q: he study focuses on optimizing window design for energy efficiency in high-rise office buildings using a combination of simulation platforms and genetic algorithms. While the topic is relevant, the research lacks a comprehensive theoretical framework that ties the findings to broader principles in building science or energy efficiency. The paper is more applied in nature, which raises questions about its contribution to scientific knowledge beyond the specific case study of Yantai.

A: In this research paper, we have innovatively proposed an energy efficient optimization method for standard floor façade window openings in high-rise buildings. Regarding the optimization method proposed in this study, it is emphasized here that: firstly, the process of using a genetic algorithm is the process of finding a more energy-efficient way of opening windows. Secondly, the fact that the object of this study focuses on ruled office buildings in the Yantai region does not affect the fact that the research method itself has some generalizability, such as window openings in buildings in other climatic zones. The above two points indicate that the purpose of this study is to make buildings energy efficient by optimizing the window opening method, and the object of the study is chosen to be ruled office buildings in the Yantai region, which is in a cold climate zone.

Q: The paper claims to present a new approach to optimizing window design, but it does not clearly demonstrate how this approach represents a significant scientific advance. The study appears to be a practical application of existing tools rather than a novel contribution to the field.

A: In this research paper, we have innovatively proposed an energy efficient optimization method for standard floor façade window openings in high-rise buildings. The design method can optimize the parameter combinations of the geometric features of the windows and subsequently achieve the effect of building energy efficiency. To a certain extent, it makes up for the research about genetic algorithm assisted building energy-saving window opening, especially in the geometric value of window opening. The Octopus plug-in containing the genetic algorithm has strong visibility and interactivity, but it is not intuitive enough in terms of the optimization search result data, i.e., it requires a lot of human resources to process the data, so the optimization design method of this study proposes a multi-objective optimization result clustering analysis, which improves the efficiency in the process of selecting the solution.

Q: The conclusions are presented without adequate support from the data. The paper makes broad claims about the effectiveness of the proposed window design strategies without fully demonstrating how the data supports these claims.

A: The conclusions of the article are obtained from the data base of single and multi-objective optimization results. The building simulation platform mentioned in this study is widely used for building design and energy simulation, and the design methodology proposed in the study can be extended to the study of building window opening topics in different climate zones.

Q: The paper does not demonstrate an adequate understanding of the relevant literature. While it references some studies, it fails to engage deeply with significant works that could provide context or challenge the findings. Important literature on building energy efficiency and genetic algorithms is either overlooked or inadequately discussed.The conclusions drawn from the results are not fully supported by the data, and there is a disconnect between the results and the broader implications of the study. The paper fails to adequately tie together its findings with the existing body of knowledge.The paper does not clearly identify implications for research, practice, or society. It fails to bridge the gap between theory and practice, and the practical applications of the research are not well articulated. The societal impact of the research is not discussed, limiting its relevance and value.

A: The design methodology proposed in this study is highly applicable and has practical implications in the early building design phase and the later building renovation phase. The conclusions of this study are based on the theoretical model, and the next research direction is that we will apply the method to actual projects and verify the conclusions of this study by comparing the measured data with the simulated data, and at the same time prove the practicability of the design method, laying the foundation for the promotion of the design method in the future. Thank you for your comments, which provide great inspiration for our future research, and we hope you will continue to pay attention to our research.

Above, are all my responses to the two reviewers, thank you again for your valuable comments.

Best regards,

Weixiang Zhang

---

## [Decision Letter · Decision Letter 3]

20 Aug 2024

HypE-GA based study on optimal design of standard floor facade windowing of high-rise office buildings facing energy saving in heating, cooling and lighting

PONE-D-24-03690R3

Dear Dr. ZHANG WEIXIANG,

We’re pleased to inform you that your manuscript has been judged scientifically suitable for publication and will be formally accepted for publication once it meets all outstanding technical requirements as well as comments from the reviewer 1.

Kind regards,

Ashfaque Ahmed Chowdhury, Ph.D., FHEA, FIEB

Academic Editor

PLOS ONE

Reviewers' comments:

Reviewer's Responses to Questions

**Comments to the Author**

1. If the authors have adequately addressed your comments raised in a previous round of review and you feel that this manuscript is now acceptable for publication, you may indicate that here to bypass the “Comments to the Author” section, enter your conflict of interest statement in the “Confidential to Editor” section, and submit your "Accept" recommendation.

Reviewer #1: All comments have been addressed

Reviewer #2: All comments have been addressed

2. Is the manuscript technically sound, and do the data support the conclusions?

Reviewer #1: Yes

Reviewer #2: Yes

3. Has the statistical analysis been performed appropriately and rigorously? 

Reviewer #1: Yes

Reviewer #2: N/A

4. Have the authors made all data underlying the findings in their manuscript fully available?

Reviewer #1: Yes

Reviewer #2: Yes

5. Is the manuscript presented in an intelligible fashion and written in standard English?

Reviewer #1: Yes

Reviewer #2: Yes

6. Review Comments to the Author

Reviewer #1: In line 74 of the revised manuscript, the sentence starts with "In the authors' previous studies, " but the following mentioned studies are not of the authors. Please remove that opening sentence.

Reviewer #2: The article has been significantly improved, and in my opinion, it is now ready for publication. Congratulations to the authors on their excellent work

7. PLOS authors have the option to publish the peer review history of their article (what does this mean?). If published, this will include your full peer review and any attached files.

Reviewer #1: No

Reviewer #2: **Yes: **Esmatullah Noorzai Lecturer, Dept. of Project and Construction Management, School of Architecture, College of Fine Arts, University of Tehran, Tehran, Iran. Postal Code: 3139643438, +98-9125396360

esmatullah.noorzai1980@gmail.com

---

## [Editor Report · Acceptance letter]

14 Oct 2024

PONE-D-24-03690R3 

PLOS ONE

Dear Dr. Zhang, 

I'm pleased to inform you that your manuscript has been deemed suitable for publication in PLOS ONE. Congratulations! Your manuscript is now being handed over to our production team.

Kind regards, 

on behalf of

Dr. Ashfaque Ahmed Chowdhury 

Academic Editor

PLOS ONE